# Black-Box Assisted Regression: Phase Transitions and Minimax Optimality

Yan Zhou [1]

## Abstract

Foundation models are often used as fixed black-box predictors for downstream tasks with limited labeled data, but their predictions may be biased and unsafe to trust blindly. We study this setting through black-box assisted nonparametric regression: a learner observes labeled samples and can query a fixed predictor $f_0$, while the target $f^*$ is close to $f_0$ in $L_2(P_X)$ up to an unknown radius $\delta$. We give a finite-sample minimax characterization showing a phase transition at $\delta_c(n) \asymp n^{-\beta/(2\beta+d)}$, with leading risk $\min\{\delta^2, n^{-2\beta/(2\beta+d)}\}$. We then analyze a Safe Residual Estimator: it learns a correction around $f_0$, initializes the residual head at zero so the initial predictor equals $f_0$, and uses holdout selection to revert to $f_0$ when the learned correction is not supported by validation data. Here, "safe" means avoiding negative transfer, i.e., performing worse than the black-box predictor alone. The estimator matches the leading minimax term up to an additive validation-selection cost. Synthetic regression experiments verify the predicted phase transition, while CIFAR-100 with CLIP and AG News with Qwen3-8B provide practice-facing evidence that the same residual-correction tradeoff is useful beyond the formal squared-loss regression setting.

## 1. Introduction

Modern machine-learning pipelines often start from a strong pretrained system whose internal parameters or source data are unavailable to the downstream learner. The learner may only query a fixed black-box predictor $f_0$ and then use a small labeled dataset from the target task. This access model differs from standard fine-tuning, transfer learning, and distillation: we do not assume source data, parameter-

---

[1] School of Mathematics and Statistics, Changsha University of Science and Technology. Correspondence to: Yan Zhou <yanzhou@stu.csust.edu.cn>.

*Proceedings of the 43rd International Conference on Machine Learning*, Seoul, South Korea. PMLR 306, 2026. Copyright 2026 by the author(s).

space access, or the ability to modify the pretrained model. The statistical question is therefore not how to train the foundation model, but how to use its predictions safely when its error on the target distribution is unknown.

We formalize this problem as *Black-Box Assisted Non-Parametric Regression*. The unknown target regression function $f^*$ is assumed to satisfy $\|f^* - f_0\|_{L_2(P_X)} \le \delta$ for an unknown prior error $\delta$. Small $\delta$ defines a prior-dominated regime in which the black-box predictor is already reliable; large $\delta$ defines a sample-dominated regime in which labeled data must drive the correction. The challenge is to adapt to these regimes without knowing $\delta$ and without suffering negative transfer, meaning a final predictor whose risk is worse than using $f_0$ alone.

Our contribution is a finite-sample learning-theoretic characterization of this tradeoff. First, we prove a minimax lower bound showing that the leading risk is governed by $\min\{\delta^2, n^{-2\beta/(2\beta+d)}\}$. Second, we analyze a deliberately simple *Safe Residual Estimator* that learns a correction $\hat{r}$ and then selects between $f_0$ and $f_0 + \hat{r}$ on held-out data. Third, we show that this estimator matches the leading minimax term up to an additive validation cost and satisfies an oracle-style no-negative-transfer guarantee. Finally, we use synthetic regression experiments to check the theorem-level predictions and vision/NLP experiments to probe the same qualitative mechanism in practical classification pipelines.

**Conflict of Interest Disclosure.** The author declares no financial conflicts of interest related to this work.

## 2. Related Work

**Transfer, adaptation, and distillation.** Classical transfer learning and domain adaptation often assume access to source data, source parameters, shared representations, or an adaptable pretrained model (Pan & Yang, 2010; Weiss et al., 2016; Cai & Pu, 2024; Lin & Reimherr, 2025; Kuzborskij & Orabona, 2013). Distillation similarly uses teacher predictions, but typically studies compression or training a student to imitate a teacher. Our setting is narrower and more statistical: the learner only receives fixed black-box predictions $f_0(x)$ on target-task inputs and seeks minimax-optimal target-risk guarantees under an unknown prior error.

**Pseudo-labeling, self-training, and noisy labels.** Pseudo-labeling and self-training are widely used to exploit weak or machine-generated labels (Lee, 2013; Xie et al., 2020; Wang et al., 2021), and several works provide theoretical guarantees under distribution or subpopulation shift (Cai et al., 2021; Joo & Klabjan, 2024). These results are complementary to ours. Our claim is not that self-training lacks theory, but that prior work does not characterize the minimax phase transition $\min\{\delta^2, n^{-2\beta/(2\beta+d)}\}$ for fixed black-box assisted regression. Noisy-label correction is also distinct: our target labels are standard noisy regression observations, not corrupted or mislabeled; the imperfect object is the fixed predictor $f_0$.

**Prediction-powered inference and model selection.** Prediction-powered inference (PPI) uses black-box predictions to construct valid confidence intervals (Angelopoulos et al., 2023; Zrnic & Candès, 2024), whereas our goal is risk-optimal function estimation. Our holdout step is closer to two-model validation/model selection: it selects between $f_0$ and $f_0 + \hat{r}$, enabling an oracle inequality that formalizes no negative transfer.

**Summary.** We formalize the "foundation model as prior" problem via non-parametric minimax theory, identifying the critical phase transition and proposing a safe residual estimator. A comprehensive discussion of additional literature, including neighboring transfer/adaptation methods (Gu et al., 2023; Lee et al., 2025), kernel methods, and PPI, is provided in Appendix A.

## 3. Problem Setup

We consider non-parametric regression with random design (Assumption 5.1) for the theoretical analysis. Random design means that the covariates are sampled i.i.d. from the target marginal distribution $P_X$; Assumption 5.1 specifies the bounded-domain and density conditions used in the proofs. Let $\mathcal{X} = [0,1]^d$ be the normalized input space. We assume the true regression function $f^* : \mathcal{X} \to \mathbb{R}$ belongs to a function class $\mathcal{F}$ (e.g., Hölder class or Sobolev class). We are given a dataset $\mathcal{D}_n = \{(x_i, y_i)\}_{i=1}^n$, where $y_i = f^*(x_i) + \epsilon_i$ and $\epsilon_i \sim \mathcal{N}(0, \sigma^2)$ are independent Gaussian noise.

We have access to a *black-box predictor* $f_0 : \mathcal{X} \to \mathbb{R}$ derived from a pre-trained model, fixed during inference. Assuming $f_0$ approximates $f^*$ with a global error bound $\|f_0 - f^*\|_{L_2(P_X)} \leq \delta$ (unknown $\delta \geq 0$), we construct an estimator $\hat{f}$ minimizing the MSE risk:

$$\mathcal{R}(\hat{f}, f^*) = \mathbb{E}\left[\|\hat{f} - f^*\|_{L_2(P_X)}^2\right]. \qquad (1)$$

**Definition 3.1** (Hölder Class). For $\beta > 0$ and $L > 0$, the Hölder class $\mathcal{H}(\beta, L)$ consists of functions $f : [0,1]^d \to \mathbb{R}$ whose derivatives up to order $\lfloor \beta \rfloor$ exist and are bounded, with the highest-order derivatives being $(\beta - \lfloor \beta \rfloor)$-Hölder

continuous with constant $L$. (See Appendix B for the precise definition.)

This class specifies the regularity of the residual $r^*$ in the minimax analysis of Section 5. Importantly, we do not assume that the black-box predictor $f_0$ itself belongs to a Hölder or Sobolev class. The regularity condition is placed on the residual $r^* = f^* - f_0$, together with the radius constraint $\|r^*\|_{L_2(P_X)} \leq \delta$. Thus $f_0$ may come from a much more general source, including a foundation model, as long as the target-task correction around it is regular.

## 4. Method

The estimator is intentionally simple. Its role is not to compete with all possible transfer-learning heuristics, but to provide a clean statistical object for which the minimax tradeoff and no-negative-transfer property can be proved. The procedure has two candidates: the black-box predictor $f_0$ and a residual-corrected predictor $f_0 + \hat{r}$. It then uses held-out data to select between them.

We call the procedure safe because the selection step is designed to avoid negative transfer: if the learned residual correction does not improve held-out loss relative to $f_0$, the final predictor reverts to $f_0$. The binary selection is deliberate. Estimating a continuous mixing weight from scarce validation data can be unstable, whereas the two-candidate selection admits the oracle inequality in Theorem 5.7.

### 4.1. The Safe Residual Estimator

The core intuition behind our method is to decompose the complexity of the learning task. We posit that the true function can be expressed as the sum of the black-box prior and a residual component:

$$f^*(x) = f_0(x) + r^*(x), \qquad (2)$$

where $r^*(x) = f^*(x) - f_0(x)$ is the target-task residual error of the black-box predictor.

To achieve adaptivity to the unknown quality of the black-box prior and mitigate negative transfer, we employ a *Select-and-Estimate* strategy. We split the available labeled data into a training set $\mathcal{D}_{\mathrm{tr}}$ and a validation set $\mathcal{D}_{\mathrm{val}}$. We train a residual model $\hat{r}$ on $\mathcal{D}_{\mathrm{tr}}$ and then decide whether to use it based on performance on $\mathcal{D}_{\mathrm{val}}$.

The residual-corrected candidate is $\hat{f}_{\mathrm{res}} = f_0 + \hat{r}$. The final Safe Residual Estimator takes the form:

$$\hat{f}_{\mathrm{safe}}(x) = f_0(x) + \hat{\alpha} \cdot \hat{r}(x), \qquad (3)$$

where $\hat{\alpha} \in \{0, 1\}$ is a selection variable. If $\hat{\alpha} = 0$, this triggers a reversion to the black-box; if $\hat{\alpha} = 1$, we correct it. This mechanism is crucial for the "black-box dominated"

regime where learning the residual might be noisier than the bias itself. The detailed procedure, including the splitting and selection logic, is formally described in Algorithm 1.

---

**Algorithm 1** Safe Residual Estimator

**Input:** Data $\mathcal{D}_{\text{tr}}, \mathcal{D}_{\text{val}}$, Prior $f_0$.
  **1. Residual Learning:**
    Obtain $\hat{r}$ by minimizing MSE of $(y - f_0(x))$ on $\mathcal{D}_{\text{tr}}$.
  **2. Risk Assessment:**
    $\mathcal{L}_{\text{BB}} \leftarrow \sum_{(x,y) \in \mathcal{D}_{\text{val}}} (y - f_0(x))^2$
    $\mathcal{L}_{\text{Res}} \leftarrow \sum_{(x,y) \in \mathcal{D}_{\text{val}}} (y - f_0(x) - \hat{r}(x))^2$
  **3. Safe Selection:**
    $\hat{\alpha} \leftarrow \mathbb{I}\left[\mathcal{L}_{\text{Res}} < \mathcal{L}_{\text{BB}}\right]$
  **Output:** $\hat{f}_{\text{safe}}(x) = f_0(x) + \hat{\alpha} \cdot \hat{r}(x)$

---

The algorithm selects between $f_0$ and $\hat{f}_{\text{res}} = f_0 + \hat{r}$; it is not selecting between zero-shot and scratch. Therefore the selected residual candidate can outperform both the black-box predictor and a scratch estimator when the correction $\hat{r}$ is estimable.

For neural residual heads used in the classification experiments, we use zero-initialization: the residual head is initialized so that $\hat{r} \equiv 0$ at the start of training. Thus the initial predictor equals $f_0$, and the model departs from the black-box prior only when the labeled data provide a training signal. Detailed hyperparameters and architectures are provided in Appendix F.

### 4.2. Why Residualization Rather Than Linear Ensembling?

Linear or validation-tuned ensembling is a natural model-selection baseline, and related forms of ensembling and weight-space adaptation are common in transfer and few-shot settings (Gu et al., 2023; Lee et al., 2025). In our fixed function-access setting, however, a convex combination of $f_0$ and a scratch estimator can only move within their linear span.

A common alternative to residual learning is the weighted ensemble, $\hat{f}_{\alpha}(x) = \alpha f_0(x) + (1 - \alpha)\hat{f}_{\text{scratch}}(x)$, where $\alpha \in [0, 1]$. While effective for variance reduction, we highlight a fundamental geometric limitation of this approach in correcting residual errors that are not aligned with the scratch estimator.

Consider the Hilbert space $L_2(P_X)$. The weighted ensemble restricts the estimator to the affine line segment connecting the prior $f_0$ and the data-driven estimator $\hat{f}_{\text{scratch}}$. Let the true residual be $r^* = f^* - f_0$. The approximation error is minimized when $(1 - \alpha)(\hat{f}_{\text{scratch}} - f_0) \approx r^*$. However, $\hat{f}_{\text{scratch}}$ is trained to approximate $f^*$, not to align with the bias direction $r^*$.

Formally, let $\mathcal{S} = \text{span}\{f_0, \hat{f}_{\text{scratch}}\}$ be the subspace spanned by the estimators. If the residual error $r^*$ contains a component in the orthogonal complement $\mathcal{S}^{\perp}$ (e.g., $r^*$ contains high-frequency corrections missing from both $f_0$ and the smooth approximation $\hat{f}_{\text{scratch}}$), the weighted ensemble is fundamentally insensitive to it:

$$\inf_{\alpha \in \mathbb{R}} \|\hat{f}_{\alpha} - f^*\|^2 \geq \|\mathcal{P}_{\mathcal{S}^{\perp}}(r^*)\|^2, \tag{4}$$

where $\mathcal{P}_{\mathcal{S}^{\perp}}$ is the orthogonal projection operator.

*Remark* 4.1 (Geometric Interpretation: Subspace vs. Ball). The fundamental limitation of the weighted ensemble lies in its geometry. The estimator $\hat{f}_{\alpha}$ is constrained to search within a one-dimensional affine subspace connecting $f_0$ and $\hat{f}_{\text{scratch}}$. In contrast, our residual estimator searches within a ball centered at $f_0$ in the function space $\mathcal{F}$. From a manifold perspective, if the bias $r^*$ lies in the null space of the projection operator onto the line spanned by $\{f_0, \hat{f}_{\text{scratch}}\}$, the weighted ensemble is blind to it. Our method, by explicitly modeling the residual map, effectively performs a *local re-centering* of the hypothesis class, allowing it to recover high-frequency corrections that are orthogonal to the base predictors.

### 4.3. Structural Advantage and Risk Decomposition

To understand why the RESIDUAL estimator provides a fundamental advantage, we analyze the decomposition of the expected risk. Let $\mathcal{R}(\hat{f}) = \mathbb{E}[\|\hat{f} - f^*\|^2]$ be the $L_2$ risk. For a standard estimator $\hat{f}_{\text{scratch}}$ trained on $\mathcal{D}_n$, the risk is governed by the complexity of the entire function class $\mathcal{F}$.

In our framework, we write $\hat{f}(x) = f_0(x) + \hat{r}(x)$. Since $f_0$ is a fixed, deterministic black-box, the risk of our estimator becomes:

$$\begin{aligned} \mathcal{R}(\hat{f}) &= \mathbb{E}_{X,\epsilon} \|(f_0 + \hat{r}) - (f_0 + r^*)\|_{L_2}^2 \\ &= \mathbb{E} \|\hat{r} - r^*\|_{L_2}^2. \end{aligned} \tag{5}$$

This identity reveals a crucial insight: the statistical difficulty of the task is shifted from estimating $f^*$ to estimating the residual $r^*$. From a bias-variance perspective, if we use a kernel-based learner with regularization parameter $\lambda$, the risk of $\hat{r}$ can be bounded by:

$$\mathcal{R}(\hat{r}) \leq \underbrace{\|(I - \mathcal{K}_{\lambda})r^*\|^2}_{\text{Approximation Bias}} + \underbrace{\frac{\sigma^2}{n}\text{tr}(\mathcal{K}_{\lambda}^2)}_{\text{Estimation Variance}}, \tag{6}$$

where $\mathcal{K}_{\lambda}$ is the smoothing operator. While the variance term remains $O(n^{-1} \cdot \text{effective dim})$, the bias term now depends only on the energy of $r^*$. Under Assumption 5.2, $\|r^*\| \leq \delta$. When $\delta$ is small, the approximation bias is significantly lower than that of $\hat{f}_{\text{scratch}}$, which must account for the full magnitude $\|f^*\|$. This structural prior effectively "centers" the learning process around $f_0$, allowing the model

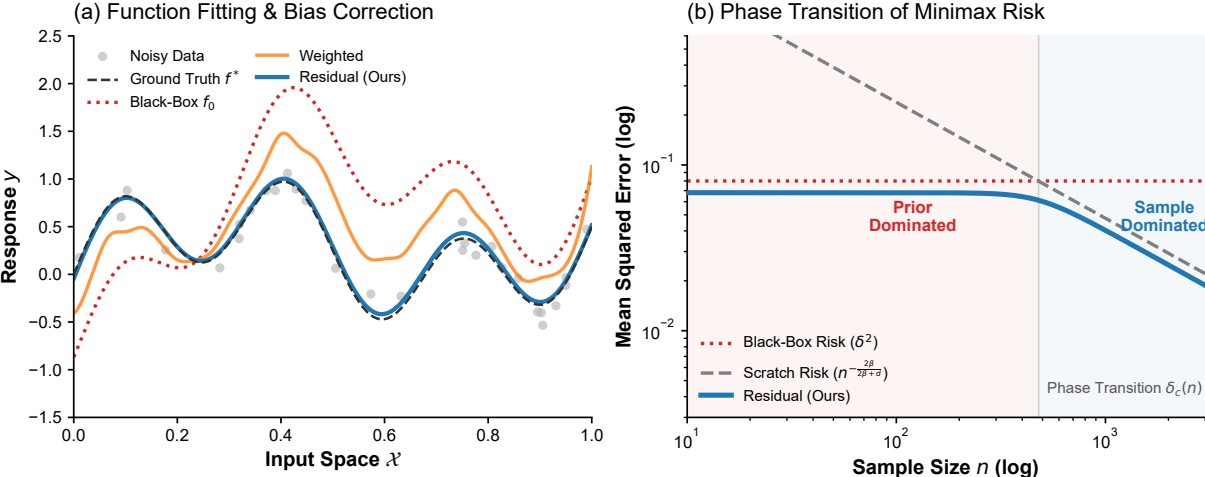

*Figure 1.* **Theory in Action: The Phase Transition. (a) Mechanism:** Unlike WEIGHTED ensembles (orange) that are constrained to a linear subspace spanned by the prior and data, our RESIDUAL estimator (blue) performs a non-linear correction, effectively capturing the complex residual $r^*(x)$. **(b) Risk Profile (Key Theoretical Result):** We identify a critical threshold $\delta_c(n)$. **Left of dashed line (Prior-Dominated):** The black-box is accurate ($\delta$ is small); our estimator adapts and achieves the fast rate $\delta^2$. **Right of dashed line (Sample-Dominated):** The black-box is biased; our estimator safely recovers the optimal non-parametric rate $n^{-\alpha}$, avoiding the high error floor of the black-box. Note how SAFE RESIDUAL (blue curve) tracks the lower envelope by selecting between $f_0$ and $f_0 + \hat{r}$, not between zero-shot and scratch.

to allocate its limited sample budget entirely to correcting the prior's deficiencies.

# 5. Theoretical Analysis

We give finite-sample minimax bounds for black-box assisted regression. The results characterize the leading minimax term and the additional cost introduced by holdout selection, rather than claiming exact equality between the lower and upper bounds. Theorem 5.4 gives the minimax lower bound and identifies the phase transition. Theorem 5.5 analyzes a validation-selected residual estimator and shows that it adapts to the unknown prior quality. Theorem 5.7 is an oracle inequality for the concrete holdout selection step in Algorithm 1, showing that the selected predictor performs almost as well as the better of $f_0$ and $f_0 + \hat{r}$.

## 5.1. Assumptions and Definitions

We consider the non-parametric regression problem $y_i = f^*(x_i) + \epsilon_i$ with $\epsilon_i \sim \mathcal{N}(0, \sigma^2)$. To rigorously relate the empirical risk to the population $L_2$ risk, we introduce a standard design assumption.

**Assumption 5.1** (Random Design). The covariates $\{x_i\}_{i=1}^n$ are i.i.d. samples from a distribution $P_X$ on $\mathcal{X} = [0,1]^d$ with Lebesgue density $p(x)$ satisfying $c \le p(x) \le C$ for constants $0 < c \le C < \infty$.

Next, we define the structure of the true function $f^*$ relative to the black-box $f_0$. Instead of assuming $f^*$ lies in a generic ball, we define the function class explicitly in terms of the

residual structure.

**Assumption 5.2** (Residual Regularity and Boundedness). Let $f_0$ be a fixed black-box predictor. The true function satisfies $f^* = f_0 + r^*$ where $r^* \in \mathcal{H}(\beta, L)$ and $\|r^*\|_{L_2(P_X)} \le \delta$. Moreover, assume $\|r^*\|_\infty \le B$ and $\|f_0\|_\infty \le B_0$.

This definition implies that the residual $r^* = f^* - f_0$ is smooth and has bounded $L_2(P_X)$ energy (controlled by $\delta$).

**Definition 5.3** (Black-box Assisted Class). Define

$$\mathcal{F}(\delta) := \{f_0 + r : r \in \mathcal{H}(\beta, L), \\ \|r\|_{L_2(P_X)} \le \delta, \ \|r\|_\infty \le B\}. \tag{7}$$

## 5.2. Minimax Lower Bounds

We first characterize the fundamental limit of this learning problem. The following theorem establishes an information-theoretic lower bound for any estimator.

**Theorem 5.4** (Minimax Lower Bound). *Under Assumptions 5.1 and 5.2, there exists a constant $c > 0$ depending only on $\beta, L, d, \sigma$ and the density bounds in Assumption 5.1 such that:*

$$\inf_{\hat{f}} \sup_{f^* \in \mathcal{F}(\delta)} \mathbb{E}\|\hat{f} - f^*\|_{L_2(P_X)}^2 \\ \ge c \cdot \min\left(\delta^2, n^{-\frac{2\beta}{2\beta+d}}\right). \tag{8}$$

*Proof sketch in Appendix C; full proof in Appendix D.2.*

This theorem rigorously establishes the **Phase Transition**. The term $\delta^2$ represents the irreducible bias floor if one relies

*Table 1.* **1D Synthetic Regression (MSE)**. Comparison of risk under varying sample size $n$ and prior error $\delta$. RES(SAFE) denotes the selected predictor that chooses between $f_0$ and $f_0 + \hat{r}$, not between BB and SCRATCH. **Key Observation:** When the black-box is perfect ($\delta = 0.0$), RESIDUAL(RAW) suffers from negative transfer (0.0335), but RESIDUAL(SAFE) effectively falls back (Fallback=65%), reducing risk to 0.0027. When bias is large ($\delta = 0.8$), our method significantly outperforms WEIGHTED (0.097 vs 0.145) by modeling the non-linear residual. In the extreme bias regime ($\delta = 1.5$), RESIDUAL (0.146) dramatically outperforms SCRATCH (0.189).

| SETTING | BB | WGT | CAT | RES(RAW) | RES(SAFE) | FALLBACK | SCRATCH |
|---|---|---|---|---|---|---|---|
| $n = 20, \delta = 0.0$ | **0.0000** | 0.0007 | 0.0501 | 0.0335 | 0.0027 | 65% | 0.1894 |
| $n = 20, \delta = 0.1$ | 0.0100 | 0.0096 | 0.0589 | 0.0368 | **0.0118** | 60% | 0.1894 |
| $n = 20, \delta = 0.8$ | 0.6399 | 0.1452 | 0.1448 | **0.0857** | 0.0971 | 5% | 0.1894 |
| $n = 20, \delta = 1.5$ | 2.2496 | 0.1700 | 0.1599 | **0.1461** | **0.1461** | 0% | 0.1894 |
| $n = 50, \delta = 0.0$ | **0.0000** | 0.0026 | 0.0107 | 0.0108 | 0.0006 | 65% | 0.0325 |
| $n = 50, \delta = 0.8$ | 0.6399 | 0.0311 | 0.0712 | **0.0130** | **0.0130** | 0% | 0.0325 |

*Table 2.* **High-Dimensional Regression** ($d = 20, n = 100$). In high dimensions with limited samples, learning the residual is difficult. SCRATCH suffers from the curse of dimensionality, with risk staying high (0.489) even at $n = 100$. RES(SAFE) selects between $f_0$ and $f_0 + \hat{r}$. When the prior is poor ($\delta = 1.0$), RESIDUAL(RAW) explodes (1.026) due to the curse of dimensionality, performing worse than the black-box (0.994). Our SAFE RESIDUAL detects this failure, triggering fallback (60%) and stabilizing the risk at 1.007.

| $\delta$ | BB | WGT | CAT | RES(RAW) | RES(SAFE) | FALLBACK | SCRATCH |
|---|---|---|---|---|---|---|---|
| 0.0 | **0.000** | 0.006 | 0.034 | 0.006 | **0.002** | 40% | 0.489 |
| 0.2 | **0.040** | 0.047 | 0.074 | 0.048 | **0.040** | 70% | 0.489 |
| 0.5 | 0.248 | 0.253 | 0.280 | 0.263 | 0.250 | 60% | 0.489 |
| 1.0 | 0.994 | **0.485** | 0.509 | 1.026 | 1.007 | 60% | 0.489 |

solely on the black-box, while $n^{-\frac{2\beta}{2\beta+d}}$ represents the optimal rate for learning the residual from data. The difficulty is governed by the minimum of these two quantities. See Appendix C for geometric intuition regarding the packing argument.

The critical threshold for this phase transition is given by:

$$\delta_c(n) \asymp n^{-\frac{\beta}{2\beta+d}}. \tag{9}$$

When $\delta \ll \delta_c(n)$, we are in the black-box dominated regime; otherwise, we enter the sample dominated regime.

### 5.3. Upper Bounds and Adaptivity

Can we achieve this lower bound? A naive estimator might fail to adapt to the unknown $\delta$. We analyze a theoretically grounded *Select-and-Estimate* procedure matching the Safe Residual Estimator: it uses sample splitting to choose between the black-box and the residual-corrected estimator.

The following bounds are non-asymptotic: they hold for every finite $n$, every $\delta \geq 0$, and every fixed black-box predictor $f_0$. In particular, $\delta$ is a property of the black-box prior and does not vary with $n$. The phase transition at $\delta_c(n) \asymp n^{-\beta/(2\beta+d)}$ should therefore be read as a finite-sample tradeoff between prior error and the difficulty of estimating the residual from labeled data.

**Theorem 5.5** (Upper Bound). *There exists an estimator $\hat{f}_{\text{safe}}$ (constructed via sample splitting, a minimax-optimal*

*nonparametric residual regressor on $\mathcal{D}_{\text{tr}}$, and safe holdout selection on $\mathcal{D}_{\text{val}}$) such that for all $\delta \geq 0$,*

$$\sup_{f^* \in \mathcal{F}(\delta)} \mathbb{E}\|\hat{f}_{\text{safe}} - f^*\|^2_{L_2(P_X)}$$
$$\leq C \cdot \min\left(\delta^2, n^{-\frac{2\beta}{2\beta+d}}\right) + \frac{C'}{n}, \tag{10}$$

*where $C$ and $C'$ depend only on $(\beta, L, d, \sigma, B, B_0, c, C)$ and not on $n$, $\delta$, or $f_0$.*

*See Appendix D.3 for the detailed construction and proof.*

**Implication.** This result shows that the estimator matches the leading minimax term up to the additive validation-selection cost $C'/n$. Since $2\beta/(2\beta + d) < 1$, this cost is asymptotically lower order than the standard nonparametric term, but it can be the active finite-sample price in the strongly prior-dominated regime.

**Corollary 5.6** (Near-Minimax Rate of the Safe Residual Estimator). *Let $\hat{f}_{\text{safe}}$ denote the output of Algorithm 1 with split ratio $\rho \in (0, 1)$. Under Assumptions 5.1–5.2, $\hat{f}_{\text{safe}}$ is minimax-optimal up to the validation-selection cost and a constant inflation factor from sample splitting:*

$$\sup_{f^* \in \mathcal{F}(\delta)} \mathbb{E}\|\hat{f}_{\text{safe}} - f^*\|^2_{L_2(P_X)}$$
$$\leq (1 - \rho)^{-\frac{2\beta}{2\beta+d}} \cdot C \min\left(\delta^2, n^{-\frac{2\beta}{2\beta+d}}\right) \tag{11}$$
$$+ \tilde{O}(n^{-1}).$$

*In particular, with $\rho = 0.2$ and $\beta \approx d$, the inflation factor is $(1 - \rho)^{-\frac{2\beta}{2\beta+d}} \approx 1.16$. Thus, Algorithm 1 matches the leading lower-bound term in Theorem 5.4 up to constant factors and the additive validation-selection cost.*

## 5.4. Theoretical Guarantee for Safe Selection

A key component of our method is the validation-based selection mechanism (Step 2 in Algorithm 1). We now give a rigorous oracle inequality for the selected predictor.

**Two notions of risk.** For any (possibly data-dependent) predictor $f$, define the *population estimation error*

$$\mathrm{Err}(f) := \|f - f^*\|_{L_2(P_X)}^2,$$

which is random because $f$ depends on the training data. Also define the *prediction risk*

$$L(f) := \mathbb{E}\big[(Y - f(X))^2 \,\big|\, f\big].$$

Under the model $Y = f^*(X) + \epsilon$ with $\mathbb{E}[\epsilon^2] = \sigma^2$, we have the exact identity

$$L(f) = \mathrm{Err}(f) + \sigma^2. \tag{12}$$

Hence, oracle guarantees for prediction risk immediately translate to estimation error, up to an additive constant $\sigma^2$.

**Setup.** Let $\hat{f}_{\mathrm{BB}} := f_0$ and $\hat{f}_{\mathrm{Res}} := f_0 + \hat{r}$, where $\hat{r}$ is trained on $\mathcal{D}_{\mathrm{tr}}$. On an independent validation set $\mathcal{D}_{\mathrm{val}}$ of size $n_{\mathrm{val}}$, define the empirical validation losses

$$\hat{L}_{\mathrm{val}}(f) := \frac{1}{n_{\mathrm{val}}} \sum_{(x,y) \in \mathcal{D}_{\mathrm{val}}} (y - f(x))^2,$$

and select $\hat{f}_{\mathrm{safe}} \in \{\hat{f}_{\mathrm{BB}}, \hat{f}_{\mathrm{Res}}\}$ by minimizing $\hat{L}_{\mathrm{val}}(\cdot)$.

**Theorem 5.7** (Oracle Inequality for Safe Selection (hold-out))**.** *Assume $\epsilon$ is sub-Gaussian with parameter $\sigma$ (Gaussian is a special case), and $\|f^*\|_\infty \leq B_0 + B$ (implied by Assumption 5.2). Fix any $\gamma \in (0,1)$ and $\delta' \in (0,1)$. Then, conditional on the training data $\mathcal{D}_{\mathrm{tr}}$, with probability at least $1 - \delta'$ over the validation sample $\mathcal{D}_{\mathrm{val}}$,*

$$
\begin{aligned}
\mathrm{Err}(\hat{f}_{\mathrm{safe}}) &\leq (1 + \gamma) \\
&\quad \cdot \min\big\{\mathrm{Err}(\hat{f}_{\mathrm{BB}}), \mathrm{Err}(\hat{f}_{\mathrm{Res}})\big\} \\
&\quad + C \frac{\log(2/\delta')}{n_{\mathrm{val}}},
\end{aligned}
\tag{13}
$$

*where $C$ depends only on $(B_0 + B, \sigma)$.*

*Proof provided in Appendix D.5.*

**Remark (no negative transfer up to validation cost).** The oracle inequality formalizes safety: relative to the two candidates considered by the algorithm, the selected predictor is within an additive validation cost of the better candidate. In particular, because $f_0$ is one of the candidates, the selected predictor cannot be substantially worse than the black-box route except for the finite-sample validation term.

**Remark (expectation form).** Integrating (13) over $\mathcal{D}_{\mathrm{val}}$ yields an expectation bound of the same form (with constants), and using (12) one may equivalently state the oracle inequality in prediction risk. A complete proof is provided in Appendix D.

## 5.5. The Price of Safety: Analysis of Sample Splitting

Our proposed estimator relies on sample splitting, using $n_{\mathrm{tr}}$ samples for learning the residual and $n_{\mathrm{val}}$ samples for selection. A natural concern is the statistical efficiency loss due to reducing the effective training set size from $n$ to $n_{\mathrm{tr}} = (1 - \rho)n$, where $\rho \in (0,1)$ is the validation fraction (e.g., $\rho = 0.2$).

We quantify this "price of safety" as the ratio of the minimax risks. In the sample-dominated regime where learning is necessary, the risk scales as $n^{-\frac{2\beta}{2\beta+d}}$. The inflation factor due to splitting is:

$$
\begin{aligned}
\text{Inflation Factor} &= \frac{((1-\rho)n)^{-\frac{2\beta}{2\beta+d}}}{n^{-\frac{2\beta}{2\beta+d}}} \\
&= (1-\rho)^{-\frac{2\beta}{2\beta+d}}.
\end{aligned}
\tag{14}
$$

For $\rho = 0.2$ and $\beta \approx d$, this factor is $\approx 1.16$, implying only a 16% risk increase.

In contrast, failing to select (i.e., blindly trusting a biased prior) incurs a risk of $\delta^2$, which can be arbitrarily larger than the optimal rate $n^{-\alpha}$ as $n \to \infty$. Thus, the sample splitting strategy trades a small, constant-factor increase in variance for robust protection against potentially unbounded bias.

## 5.6. Proof Intuition: Metric Entropy and Adaptivity

The convergence rates in Theorems 5.4 and 5.5 are deeply tied to the *metric entropy* of the function classes. Let $H(\epsilon, \mathcal{F}, \|\cdot\|)$ be the $\epsilon$-entropy of class $\mathcal{F}$. The minimax rate for a class is typically determined by the solution to the equation $\epsilon^2 \asymp H(\epsilon, \mathcal{F})/n$.

For the standard Hölder class $\mathcal{H}(\beta, L)$, the entropy scales as $\epsilon^{-d/\beta}$. However, our black-box assisted class $\mathcal{F}(\delta)$ (Definition 5.3) is the intersection of a smoothness ball and an $L_2$ ball of radius $\delta$.

When $\epsilon > \delta$, the $L_2$ constraint is active, and the effective entropy is drastically reduced, leading to the $\delta^2$ floor. When

$\epsilon < \delta$, the smoothness constraint dominates, recovering the $n^{-\frac{2\beta}{2\beta+d}}$ rate.

Our Safe Residual Estimator achieves this regime selection by implicitly performing a multi-scale analysis. The validation step (Algorithm 1) acts as a statistical test that checks whether the data-driven correction $\hat{r}$ has a higher signal-to-noise ratio than the prior's error $\delta$. This ensures that we only "pay" the metric entropy cost of the full Hölder class when the evidence from $\mathcal{D}_n$ is strong enough to guarantee an improvement over $f_0$.

# 6. Experiments

**Connection to Theoretical Analysis.** The experiments are designed to test the statistical mechanisms in the theory rather than to provide an exhaustive benchmark against all transfer-learning methods. Each baseline corresponds to a quantity or alternative in the analysis: BB-ONLY realizes the prior-error floor $\delta^2$; SCRATCH realizes the standard non-parametric learning route; validation-tuned WGT tests the linear-combination alternative; CONCAT tests a simple way to expose the learner to black-box information without residual centering; and RESIDUAL tests the estimator predicted by the theory.

**Synthetic Regression.** This controlled setting satisfies the squared-loss regression assumptions directly and numerically verifies the finite-sample phase transition.

**Real-World Classification (Vision & NLP).** Our formal guarantees are for squared-loss regression. The CIFAR-100 and AG News experiments use cross-entropy training and accuracy-based selection, and should not be read as theorem-level extensions of Theorems 5.4–5.7. They provide practice-facing evidence that the same qualitative trade-off between prior quality and correction complexity appears in classification when viewed at the level of score or logit functions.

For the classification experiments, we adopt the following mapping to connect with our theoretical framework: we treat multi-class classification as a regression problem on logit/score functions. Specifically, for a $K$-class problem, we learn $K$ score functions $\{f_k\}_{k=1}^{K}$, where each $f_k : \mathcal{X} \to \mathbb{R}$ predicts the logit for class $k$. The black-box prior provides initial logits $\{f_{0,k}\}_{k=1}^{K}$, and our residual estimator learns corrections $\{r_k\}_{k=1}^{K}$ such that $f_k^* = f_{0,k} + r_k^*$. The same structural intuition may remain informative, but our theorems do not prove minimax rates for cross-entropy training or accuracy-based selection.

Recent adaptation methods such as continual/class-incremental approaches or weight-space ensembling address neighboring but different access models, often requiring parameter or weight-space access to the pretrained model

(Gu et al., 2023; Lee et al., 2025). We therefore discuss them as related work rather than direct baselines for our fixed function-access formulation. All reported classification numbers are mean $\pm$ standard deviation over three independent runs; no best-run selection is used.

We compare five estimators: (1) SCRATCH: Standard supervised learning (KRR/MLP) trained solely on labeled data. (2) BB-ONLY: Direct zero-shot/few-shot predictions. (3) **WGT (Val-Tuned)**: An ensemble $\alpha f_0(x) + (1 - \alpha)\hat{f}_{scratch}(x)$, where $\alpha$ is optimized on the validation set to ensure a strong baseline. (4) CONCAT: Training on concatenated features $[x, f_0(x)]$. (5) **RESIDUAL (Ours)**: The proposed method learning $y - f_0(x)$ with zero-initialization and safe selection.

**Raw vs. Safe Residual.** We distinguish RESIDUAL (RAW) that always predicts $f_0(x) + \hat{r}(x)$, from our final SAFE RESIDUAL predictor that performs holdout selection between $\{f_0, \ f_0 + \hat{r}\}$ as in Algorithm 1. Only the latter enjoys the (no-negative-transfer) oracle guarantee in Theorem 5.7. The final method does not choose between zero-shot and scratch. It chooses between $f_0$ and the residual-corrected predictor $f_0 + \hat{r}$. Therefore, when $\hat{r}$ captures a target-task correction, the selected predictor can outperform both the black-box prior and a scratch estimator.

## 6.1. Synthetic Validation

**1D Phase Transition & Negative Transfer.** We generated data where the black-box $f_0$ has a low-frequency residual error. Figure 1 illustrates the mechanism and risk profile, while Table 1 presents the Mean Squared Error (MSE). Table 1 reports raw residual to illustrate the learning curve; our final method uses safe selection (Alg. 1) with an oracle guarantee (Thm. 5.7). In the "bad black-box" regime ($n = 20, \delta = 1.5$), BB-ONLY incurs huge error (2.25), but RESIDUAL (RAW) adapts effectively (0.146), outperforming SCRATCH (0.189). In the "small sample" regime ($n = 50, \delta = 0.0$), SAFE RESIDUAL achieves near-zero error (0.0006), vastly outperforming SCRATCH (0.0325). Notably, at $n = 20, \delta = 0.8$, SAFE RESIDUAL (0.097) beats WEIGHTED (0.145), showing the benefit of non-linear correction.

Table 1 explicitly demonstrates the protection mechanism. In the noise-dominated regime ($n = 20, \delta = 0.0$), the raw residual estimator overfits the noise (MSE 0.0335). However, the validation-based selection triggers a fallback in 65

**Breaking the Curse of Dimensionality (High-Dim).** In the $d = 20$ setting (Table 2), SCRATCH suffers from the curse of dimensionality, with risk staying high ($\sim 0.48$) at $n = 100$. In contrast, when the black-box is reasonably good ($\delta = 0.0$), SAFE RESIDUAL achieves a risk of **0.002**,

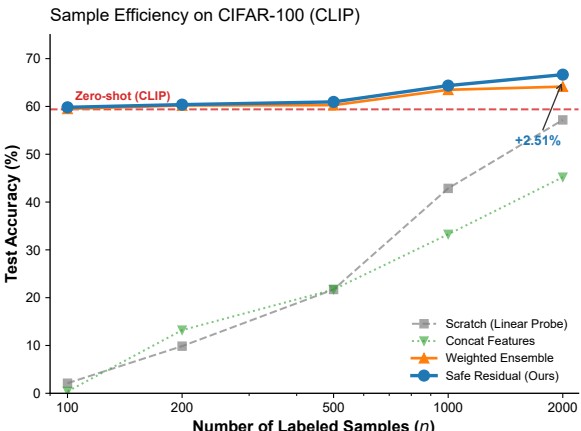

*Figure 2.* **Sample Efficiency on CIFAR-100.** Comparison of test accuracy as the number of labeled samples $n$ increases. SCRATCH (gray) and CONCAT (green) degrade severely in the few-shot regime. In contrast, our SAFE RESIDUAL estimator (blue) consistently outperforms the strong WEIGHTED ensemble baseline (orange). Notably, at $n = 2000$, our method achieves a **2.5%** gain over the weighted ensemble.

*Table 3.* **CIFAR-100 Accuracy (%)** (mean $\pm$ std over 3 runs). BB (CLIP Zero-shot) is 59.39%. Our RES method consistently outperforms baselines. Notably, at $n = 2000$, it surpasses the strong WGT (Validation-Tuned) ensemble by **2.7%**. Zero-initialization provided implicit regularization, preventing negative transfer (Fallback=0.0% across all settings).

| $n$ | SCRATCH | CAT | WGT(VAL) | RES(SAFE) |
|-----|---------|-----|----------|-----------|
| 100 | 2.1$\pm$0.3 | 0.4$\pm$0.1 | 59.3$\pm$0.2 | **59.8$\pm$0.2** |
| 200 | 9.9$\pm$0.5 | 13.2$\pm$0.6 | 54.5$\pm$0.3 | **60.4$\pm$0.2** |
| 500 | 21.7$\pm$0.8 | 21.7$\pm$0.9 | 59.4$\pm$0.3 | **61.0$\pm$0.3** |
| 1000 | 42.9$\pm$1.2 | 33.3$\pm$1.5 | 61.8$\pm$0.4 | **64.4$\pm$0.4** |
| 2000 | 57.2$\pm$1.5 | 45.2$\pm$1.8 | 63.9$\pm$0.5 | **66.7$\pm$0.5** |

a **250$\times$ improvement**. Table 2 reports raw residual to illustrate the high-dimensional variance barrier; our final method uses safe selection (Alg. 1) with an oracle guarantee (Thm. 5.7).

In high dimensions ($d = 20$), the phase transition is delayed. As shown in Table 2 ($\delta = 1.0$), the sample size $n = 100$ is insufficient to resolve the residual, causing the raw estimator to degrade (1.026) compared to the black-box (0.994). The safe selection mechanism acts as a critical brake, maintaining performance near the black-box baseline.

### 6.2. Vision: CLIP on CIFAR-100

**Comparison with Baselines.** The results are summarized in Table 3 and visualized in Figure 2. At $n = 2000$, RES reaches **66.7%**, significantly outperforming WGT (63.9%) and CAT (45.2%). RES consistently beats WGT across all sizes, providing practice-facing evidence for the benefit of non-linear correction. **Addressing Baseline Instability:** At

*Table 4.* **NLP Task (AG News with Qwen3-8B, Accuracy %)** (mean $\pm$ std over 3 runs). BB baseline is 74.06%. At extremely low sample sizes ($n = 64$), the WGT (Val-Tuned) ensemble collapses to the scratch performance (54.7%) due to validation noise, while RES remains robust (74.7%). Fallback=0.0% across all settings.

| $n$ | SCRATCH | WGT(VAL) | CAT | RES(SAFE) |
|-----|---------|----------|-----|-----------|
| 16 | 27.4$\pm$1.5 | 71.2$\pm$0.8 | 31.0$\pm$1.8 | **75.2$\pm$0.6** |
| 32 | 29.3$\pm$1.6 | 74.5$\pm$0.7 | 34.5$\pm$1.7 | **74.1$\pm$0.5** |
| 64 | 54.7$\pm$2.0 | 54.7$\pm$0.7 | 48.8$\pm$2.2 | **74.7$\pm$0.5** |
| 128 | 64.5$\pm$1.8 | 72.7$\pm$0.6 | 65.1$\pm$1.9 | **76.6$\pm$0.7** |
| 256 | 69.7$\pm$1.5 | 75.6$\pm$0.5 | 72.3$\pm$1.6 | **77.0$\pm$0.6** |

$n = 200$, we observe that WGT underperforms the zero-shot baseline (54.5% vs 59.4%). This occurs because the validation set is too small to reliably estimate the optimal mixing weight $\alpha$, leading to overfitting. Similarly, CAT fails completely (0.4%) due to the curse of dimensionality. Our RES method avoids these pitfalls through zero-initialization. Furthermore, consistent with our $L_2$ theory, RES achieves the lowest Brier Score (MSE) of **0.443** at $n = 2000$ (vs 0.538 for Black-Box).

**Robustness in Few-Shot.** With only 100 samples (1 per class), SAFE RESIDUAL maintains performance (59.82%) slightly above the Zero-shot baseline (59.39%), successfully avoiding the collapse seen in CONCAT (0.39%) and SCRATCH (2.11%).

**Ablation Studies.** We rigorously analyze the sensitivity of our method to the validation fraction $\rho$ and the impact of zero-initialization in **Appendix G**. Figures 3a and 3b demonstrate that our method remains stable for $\rho \in [0.15, 0.3]$ and zero-initialization significantly reduces early training variance compared to random initialization.

### 6.3. NLP: Qwen3-8B on AG News

We extend our evaluation to NLP using Qwen3-8B (Yang et al., 2025). We extract 4096-dimensional feature embeddings and use prompt-based generation for $f_0$. The results are summarized in Table 4.

**Robustness in Ultra-Low Sample Regime.** The NLP task demonstrates a key advantage of our residual structure in the extreme few-shot setting ($n = 16$). As shown in Table 4, the black-box prior provides a strong baseline (74.06%). Critically, in the ultra-low sample regime, the WGT ensemble suffers from *negative transfer* (71.21% < 74.06%) due to variance in weight estimation—the limited data is insufficient to reliably estimate the optimal mixing weight. In contrast, our RES estimator achieves **75.23%** at $n = 16$, demonstrating that the residual structure with zero-initialization is more robust than linear ensembling when data is scarce.

**Analysis of Baseline Instability.** Interestingly, we observe that the validation-tuned weighted ensemble (WGT) exhibits instability in low-sample regimes (e.g., $n = 64$ in Table 4). Due to the small validation set size, the selection of $\alpha$ becomes noisy, occasionally causing the model to collapse to the weaker scratch learner (54.7% vs 74.7% for Residual). In contrast, our **Residual** estimator, regularized by zero-initialization, provides more stable performance in this fixed black-box access setting.

**Scaling and Crossover.** At $n = 256$, the gap narrows but remains significant (WGT 75.59% vs. RES 76.95%). This suggests that residual learning is most useful in few-shot regimes via inductive bias (zero-initialization). The fallback rate remains 0.0% across all settings, indicating that zero-initialization provides implicit regularization even in the $n = 16$ regime.

## 7. Discussion

**Statistical Interpretation.** From a decision-theoretic perspective, the black-box $f_0$ centers the hypothesis space. Our residual estimator restricts the search to a local ball $\mathcal{B}(f_0, \delta)$, reducing metric entropy and breaking the curse of dimensionality.

**Scope of the formal guarantees.** The theorems in this paper are for squared-loss regression under the stated random-design and residual-regularity assumptions. The classification experiments are meant to probe the same residual-correction mechanism in practical settings, not to establish minimax guarantees for cross-entropy loss or accuracy-based model selection.

**Design assumptions.** The compact support and bounded-density assumptions are used to obtain clean norm equivalences and packing arguments. They are standard in nonparametric random-design theory, but relaxing them to unbounded or heavy-tailed covariate distributions is an important direction for future work.

**Selection cost.** The additive $O(1/n)$ term is the finite-sample cost of holdout-based selection. It is lower order than the nonparametric rate when $d < \infty$, but can dominate in a strongly prior-dominated regime. Removing or reducing this term may require a different adaptation mechanism, such as a Lepski-type method without sample splitting, or a sharper validation analysis.

**Generative models.** The same black-box-prior plus residual-correction idea may be relevant to score or denoising-function estimation in diffusion models, but our current results do not establish such guarantees. We leave this extension to future work.

**Limitations.** Our current analysis assumes homoscedastic Gaussian noise and a static black-box. Extending the framework to heavy-tailed noise (e.g., via Huber loss) and online learning settings remains an open problem.

**Explicit vs. Implicit Safety.** Our experiments reveal two distinct mechanisms for safety. In high-noise synthetic settings (Table 1, 2), the *explicit* validation-based fallback is active (Fallback $> 0$), effectively filtering out noisy updates. In real-world tasks (Vision/NLP), where pre-trained representations are robust, we observe *implicit* safety: the fallback rate is 0.0, yet no negative transfer occurs (e.g., Table 4, $n = 16$). This is attributed to our zero-initialization strategy, which places the estimator exactly at $f_0$ at the start of training. This strong inductive bias ensures that the model only deviates from the prior when the data signal is sufficiently strong, effectively achieving "soft" safety without triggering the hard fallback switch.

## 8. Conclusion

We formulated black-box assisted regression as a finite-sample minimax problem and characterized a phase transition at $\delta_c(n) \asymp n^{-\beta/(2\beta+d)}$. The Safe Residual Estimator matches the leading minimax term up to an additive validation-selection cost and satisfies an oracle-style safety guarantee relative to the black-box predictor. Synthetic regression experiments verify the predicted phase transition, while vision and NLP experiments suggest that residual correction with zero-initialization is a useful practical principle beyond the formal regression setting.

## Acknowledgements

The author would like to thank the School of Mathematics and Statistics, Changsha University of Science and Technology, for its support.

## Impact Statement

This paper presents work whose goal is to advance the field of Machine Learning. There are many potential societal consequences of our work, none of which we feel must be specifically highlighted here.

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

# A. Extended Related Work Discussion

This appendix provides a comprehensive review of related literature organized by theme.

## A.1. Prediction-Powered Inference and Post-Prediction Inference

Prediction-Powered Inference (PPI) provides a principled framework for constructing confidence intervals by combining a small labeled dataset with a large pool of machine predictions, without assuming the predictor is correct (Angelopoulos et al., 2023). Subsequent work has sharpened the computational and statistical properties of this paradigm, including adaptive and more efficient variants (PPI++) (Angelopoulos et al., 2024), cross-fitting style constructions (Zrnic & Candès, 2024), and Bayesian-regularized extensions (Hofer et al., 2024; Cortinovis & Caron, 2025). Recent analyses further expand this scope to stratified evaluations (Fisch et al., 2024), settings with imputed covariates (Kluger et al., 2025), and non-asymptotic limits (Mani et al., 2025).

## A.2. Transfer Learning, Domain Adaptation, and Black-Box Regimes

Classical transfer learning theory typically assumes access to source data, shared representations, or parametric model classes (Cai & Pu, 2024; Lin & Reimherr, 2025). For broader context, we refer to foundational surveys on transfer learning (Pan & Yang, 2010; Weiss et al., 2016). In parallel, hypothesis-transfer and source-free domain adaptation study regimes where source data are inaccessible (Kuzborskij & Orabona, 2013; Liang et al., 2020). Recent few-shot adaptation methods also study continual or class-incremental settings, or perform adaptation in parameter or weight space (Gu et al., 2023; Lee et al., 2025). These are relevant neighboring empirical directions, but they assume an access model different from ours. Our analysis treats the pretrained system as a fixed function-access black box: the learner cannot access source data or model parameters, only predictions.

## A.3. Pseudo-Labeling, Self-Training, and Test-Time Adaptation

A large body of empirical practice combines weak/cheap labels with a small amount of gold supervision, dating back to early work on word sense disambiguation (Yarowsky, 1995). Modern approaches include pseudo-labeling and self-training for deep networks (Lee, 2013; Xie et al., 2020; Sohn et al., 2020). Several works provide theoretical guarantees under distribution or subpopulation shift (Cai et al., 2021; Joo & Klabjan, 2024); our distinction is therefore not the absence of theory in self-training, but the different statistical object: minimax risk for fixed black-box assisted regression and the phase transition induced by the prior error $\delta$. Recent surveys consolidate this area (Amini et al., 2025; Kage et al., 2025). Test-time adaptation (TTA) further explores adjusting models under shift (Liang et al., 2025), with algorithms such as entropy-minimization (Wang et al., 2021) and continual adaptation (Wang et al., 2022).

## A.4. Noisy-Label Correction

Noisy-label correction studies how to learn when the observed labels are corrupted or biased. Our setting is different: the labels are standard noisy regression observations (true signal plus i.i.d. noise), not corrupted or mislabeled; the imperfect object is a fixed black-box predictor $f_0$. Thus the statistical question is not label-noise correction, but whether and when a black-box prediction can be safely used as a prior.

## A.5. Model Selection View: Holdout, Cross-Validation, and Oracle Inequalities

Our "safe selection" step is a two-model holdout model selection procedure. This connects to the classical cross-validation literature (Arlot & Celisse, 2010) and concentration-based analyses for model selection (Massart, 2007).

## A.6. Residual Connections and Zero-Initialization in Deep Networks

Our residualization principle mirrors the residual learning perspective popularized by ResNets (He et al., 2016). For the specific inductive bias "start exactly at the black-box" (zero-initialization), we note the close relationship to deep residual training without normalization (Zhang et al., 2019). General foundations of deep learning optimization are covered in Goodfellow et al. (2016).

*Table 5.* Notation used in the theoretical analysis.

| Symbol | Meaning |
|---|---|
| $f_0$ | fixed black-box predictor |
| $f^*$ | target regression function |
| $r^* = f^* - f_0$ | target residual |
| $\hat{r}$ | learned residual estimator |
| $\hat{f}_{\text{res}} = f_0 + \hat{r}$ | residual-corrected candidate |
| $\hat{f}_{\text{safe}}$ | final Safe Residual Estimator output |
| $\delta = \|f^* - f_0\|_{L_2(P_X)}$ | black-box prior error radius |
| $\rho$ | validation fraction |

### A.7. Kernel Ridge Regression and Approximation Theory

When the residual is learned by kernel methods, sharp rates are classically studied in Caponnetto & De Vito (2007). For broader statistical learning foundations, see Hastie et al. (2001) and Schölkopf & Smola (2001). Regarding the approximation power of our function classes, recent work establishes optimal rates for deep ReLU networks in Sobolev and Besov spaces (Siegel, 2023; Yang, 2025) and analyzes Transformers as nonparametric learners (Kim et al., 2024).

### A.8. Datasets and Foundation Models

For CIFAR-100, we cite the original technical report (Krizhevsky & Hinton, 2009). For the language model backbones used in our NLP experiments, we utilize the Qwen model family (Bai et al., 2023). Local polynomial methods (Fan, 1996) serve as a canonical nonparametric reference.

## B. Notation and Basic Preliminaries

This appendix collects notation and basic facts used throughout the proofs in Appendix D. We work under Assumptions 5.1–5.2 unless stated otherwise.

### B.1. Notation

**Data and distributions.** We observe $(x_i, y_i)_{i=1}^n$ with $x_i \overset{iid}{\sim} P_X$ on $\mathcal{X} = [0,1]^d$ and

$$y_i = f^*(x_i) + \epsilon_i, \qquad \epsilon_i \overset{iid}{\sim} \mathcal{N}(0, \sigma^2), \tag{15}$$

independent of $x_i$. Assumption 5.1 states that $P_X$ has a density $p$ w.r.t. Lebesgue measure on $[0,1]^d$ satisfying

$$0 < c \le p(x) \le C < \infty \quad \text{for all } x \in [0,1]^d. \tag{16}$$

**Black-box predictor and residual.** The learner has access to a fixed black-box predictor $f_0 : \mathcal{X} \to \mathbb{R}$. We define the residual

$$r^*(x) := f^*(x) - f_0(x). \tag{17}$$

Assumption 5.2 posits $r^* \in \mathcal{H}(\beta, L)$ and $\|r^*\|_{L_2(P_X)} \le \delta$, along with uniform boundedness $\|r^*\|_\infty \le B$ and $\|f_0\|_\infty \le B_0$.

**Norms.** For a measurable function $g : \mathcal{X} \to \mathbb{R}$, define

$$\|g\|_{L_2(P_X)}^2 := \mathbb{E}_{X \sim P_X}[g(X)^2], \qquad \|g\|_{L_2}^2 := \int_{[0,1]^d} g(x)^2 \, dx, \qquad \|g\|_\infty := \sup_{x \in \mathcal{X}} |g(x)|. \tag{18}$$

When there is no ambiguity, we write $\|g\|_2 := \|g\|_{L_2(P_X)}$.

**Risk.** For an estimator $\hat{f}$ (measurable w.r.t. the data), the squared $L_2(P_X)$ risk is

$$\mathcal{R}(\hat{f}, f^*) := \mathbb{E}\big[\|\hat{f} - f^*\|_{L_2(P_X)}^2\big], \tag{19}$$

where the expectation is over the sample $(x_i, y_i)_{i=1}^n$ and any estimator randomness.

**Sample splitting.** When we split the labeled dataset into a training set $\mathcal{D}_{\mathrm{tr}}$ and a validation set $\mathcal{D}_{\mathrm{val}}$ of size $n_{\mathrm{val}}$, we write $n_{\mathrm{tr}} + n_{\mathrm{val}} = n$ and often parameterize $n_{\mathrm{val}} = \rho n$ with $\rho \in (0, 1)$.

### B.2. Norm Equivalences Under Random Design

The density bounds (16) imply that $L_2(P_X)$ and Lebesgue $L_2$ norms are equivalent on $[0, 1]^d$ up to constants depending only on $(c, C)$.

**Lemma B.1** (Equivalence of $L_2(P_X)$ and Lebesgue $L_2$)**.** *Assume* (16)*. Then for any measurable* $g : [0, 1]^d \to \mathbb{R}$,

$$c \|g\|_{L_2}^2 \leq \|g\|_{L_2(P_X)}^2 \leq C \|g\|_{L_2}^2. \tag{20}$$

*In particular,* $\|g\|_{L_2(P_X)} \asymp \|g\|_{L_2}$ *with constants depending only on* $(c, C)$.

*Proof.* By definition, $\|g\|_{L_2(P_X)}^2 = \int g(x)^2 p(x)\, dx$. Using $c \leq p(x) \leq C$ yields $c \int g(x)^2 dx \leq \int g(x)^2 p(x) dx \leq C \int g(x)^2 dx$, which is (20). $\qquad\square$

### B.3. Residual Class and Centering

Recall Definition 5.3:

$$\mathcal{F}(\delta) := \left\{ f_0 + r : r \in \mathcal{H}(\beta, L),\ \|r\|_{L_2(P_X)} \leq \delta,\ \|r\|_\infty \leq B \right\}. \tag{21}$$

The bounded-error black-box model can be viewed as a localization of the target around $f_0$ in $L_2(P_X)$.

**Lemma B.2** (Risk equivalence under residualization)**.** *For any estimator* $\hat{f}$ *and the associated residual estimator* $\hat{r} := \hat{f} - f_0$, *we have*

$$\|\hat{f} - f^*\|_{L_2(P_X)}^2 = \|\hat{r} - r^*\|_{L_2(P_X)}^2, \tag{22}$$

*and hence* $\mathcal{R}(\hat{f}, f^*) = \mathcal{R}(\hat{r}, r^*)$.

*Proof.* Since $f^* = f_0 + r^*$ and $\hat{f} = f_0 + \hat{r}$, we have $\hat{f} - f^* = \hat{r} - r^*$ pointwise. Taking $\|\cdot\|_{L_2(P_X)}^2$ yields the identity, and taking expectation over data yields the equality of risks. $\qquad\square$

### B.4. Validation Losses Used by Safe Selection

For completeness, we record the empirical validation losses used in Algorithm 1. Given a validation set $\mathcal{D}_{\mathrm{val}} = \{(x_i, y_i)\}_{i \in \mathcal{I}_{\mathrm{val}}}$,

$$\mathcal{L}_{\mathrm{BB}} := \sum_{i \in \mathcal{I}_{\mathrm{val}}} \left( y_i - f_0(x_i) \right)^2, \qquad \mathcal{L}_{\mathrm{Res}} := \sum_{i \in \mathcal{I}_{\mathrm{val}}} \left( y_i - (f_0(x_i) + \hat{r}(x_i)) \right)^2. \tag{23}$$

The selection variable is $\hat{\alpha} = \mathbf{1}\{\mathcal{L}_{\mathrm{Res}} < \mathcal{L}_{\mathrm{BB}}\}$, yielding the final predictor $\hat{f}_{\mathrm{safe}}(x) = f_0(x) + \hat{\alpha}\, \hat{r}(x)$.

## C. Geometric Interpretation and Proof Sketch

### C.1. Geometric Interpretation of the Phase Transition

The phase transition phenomenon identified in Theorem 5.4 can be understood geometrically as an intersection of function classes. The learner knows that $f^*$ lies in the intersection of the smoothness ball $\mathcal{H}(\beta, L)$ and the proximity ball $\mathcal{B}(f_0, \delta) = \{f : \|f - f_0\| \leq \delta\}$.

- **Sample Dominated Regime** ($\delta > \delta_c(n)$)**:** When the prior is relatively poor, the proximity ball $\mathcal{B}(f_0, \delta)$ is large and effectively contains the entire uncertainty region of the non-parametric estimator. In this case, the constraint $\|f - f_0\| \leq \delta$ provides little information gain over the smoothness constraint alone. The minimax risk is thus governed by the metric entropy of $\mathcal{H}(\beta, L)$, yielding the standard non-parametric rate $n^{-\frac{2\beta}{2\beta+d}}$.

- **Prior Dominated Regime** ($\delta < \delta_c(n)$)**:** Conversely, when $\delta$ is small, the proximity ball is significantly smaller than the estimation uncertainty of learning from scratch. The intersection $\mathcal{H}(\beta, L) \cap \mathcal{B}(f_0, \delta)$ is effectively constrained by the radius $\delta$. In this regime, the local complexity of the function class around $f_0$ is low, and the risk is saturated by the approximation error $\delta^2$.

The critical radius $\delta_c(n) \asymp n^{-\frac{\beta}{2\beta+d}}$ represents the boundary where the "resolution" of the available data matches the "quality" of the prior. Intuitively, with $n$ samples, the non-parametric estimator can only resolve features of scale larger than $n^{-1/(2\beta+d)}$. If the black-box error $\delta$ operates at a finer scale than this resolution limit, the data cannot improve upon the prior. The estimator only begins to improve upon $f_0$ when the sample size is large enough to resolve the fine-grained structure of the residual bias.

## C.2. Proof Sketch and Intuition

While the rigorous proofs are deferred to Appendix D, we provide here the intuition behind the phase transition established in Theorem 5.4. The minimax risk is governed by a trade-off between two sources of error: the *approximation error* inherited from the black-box and the *estimation error* associated with learning from finite samples.

**The Packing Argument.** To derive the lower bound, we employ Fano's method. We construct a "hardest" sub-problem by packing multiple hypotheses into the function class $\mathcal{F}(\delta)$. These hypotheses are local perturbations of the black-box $f_0$. Specifically, we define perturbations $\psi_j$ supported on disjoint hypercubes. The amplitude $h$ of these perturbations is constrained by two factors:

1. **Smoothness Constraint:** To remain in the Hölder class $\mathcal{H}(\beta, L)$, the amplitude must decay as the grid size decreases: $h \lesssim m^{-\beta}$.

2. **Black-Box Constraint:** To satisfy $\|f - f_0\| \leq \delta$, the aggregate energy of the perturbations cannot exceed $\delta^2$. This implies $h \lesssim \delta$.

The interplay between these two constraints creates the phase transition.

# D. Detailed Theoretical Analysis

In this appendix, we provide complete proofs for Theorems 5.4, 5.5, and 5.7. Throughout, we work under Assumptions 5.1 and 5.2 unless stated otherwise. Appendix B collects notation and basic identities used below.

## D.1. Technical Preliminaries

**Hölder class.** We use a standard (isotropic) Hölder class definition on $\mathcal{X} = [0,1]^d$. Let $\beta > 0$, write $s := \lfloor\beta\rfloor$ and $\alpha := \beta - s \in [0,1)$. For a multi-index $k = (k_1, \ldots, k_d) \in \mathbb{N}^d$ with $|k| := \sum_{j=1}^{d} k_j$, denote $\partial^k := \frac{\partial^{|k|}}{\partial x_1^{k_1} \cdots \partial x_d^{k_d}}$. Define the Hölder seminorm of order $\beta$ by

$$[f]_{\mathcal{H}(\beta)} := \max_{|k|=s} \sup_{x \neq x'} \frac{|\partial^k f(x) - \partial^k f(x')|}{\|x - x'\|_2^{\alpha}}, \tag{24}$$

with the convention that when $s = 0$, the maximum over $|k| = 0$ is interpreted as $[f]_{\mathcal{H}(\beta)} = \sup_{x \neq x'} |f(x) - f(x')|/\|x - x'\|_2^{\beta}$. Then the (ball) Hölder class $\mathcal{H}(\beta, L)$ consists of functions $f$ such that $\max_{|k|\leq s} \|\partial^k f\|_\infty \leq L$ and $[f]_{\mathcal{H}(\beta)} \leq L$.

**Probability model and the induced distributions.** For any measurable regression function $g : \mathcal{X} \to \mathbb{R}$, let $P_g$ denote the joint law of $(X_{1:n}, Y_{1:n})$ under $Y_i = g(X_i) + \epsilon_i$ with $X_i \overset{iid}{\sim} P_X$ and $\epsilon_i \overset{iid}{\sim} \mathcal{N}(0, \sigma^2)$ independent of $X_{1:n}$.

### D.1.1. KL DIVERGENCE UNDER RANDOM DESIGN

**Definition D.1** (Kullback–Leibler divergence). For probability measures $P, Q$ with $P \ll Q$, the KL divergence is $\mathrm{KL}(P\|Q) := \int \log\left(\frac{dP}{dQ}\right) dP$.

**Lemma D.2** (KL for Gaussian regression with random design). *For any measurable $g, h : \mathcal{X} \to \mathbb{R}$,*

$$\mathrm{KL}(P_g\|P_h) = \frac{n}{2\sigma^2} \|g - h\|_{L_2(P_X)}^2. \tag{25}$$

*Proof.* Condition on $X_{1:n}$. Under $P_g$, the conditional law of $Y_{1:n}$ given $X_{1:n}$ is $\mathcal{N}(\mu_g, \sigma^2 I_n)$ with $(\mu_g)_i = g(X_i)$; similarly

under $P_h$ it is $\mathcal{N}(\mu_h, \sigma^2 I_n)$ with $(\mu_h)_i = h(X_i)$. Thus,

$$\mathrm{KL}(P_g(Y_{1:n} \mid X_{1:n}) \parallel P_h(Y_{1:n} \mid X_{1:n})) = \frac{1}{2\sigma^2}\|\mu_g - \mu_h\|_2^2 = \frac{1}{2\sigma^2}\sum_{i=1}^n (g(X_i) - h(X_i))^2.$$

Since the $X$-marginal is the same under $P_g$ and $P_h$, the chain rule for KL yields

$$\mathrm{KL}(P_g \| P_h) = \mathbb{E}_{X_{1:n}}\left[\mathrm{KL}(P_g(Y_{1:n} \mid X_{1:n}) \parallel P_h(Y_{1:n} \mid X_{1:n}))\right] = \frac{1}{2\sigma^2}\sum_{i=1}^n \mathbb{E}\left[(g(X_i) - h(X_i))^2\right].$$

Using i.i.d. $X_i \sim P_X$, we get (25). $\qquad\square$

### D.1.2. VARSHAMOV–GILBERT AND FANO

**Lemma D.3** (Varshamov–Gilbert bound (Tsybakov, 2009)). *Let $\Omega = \{0,1\}^M$ with Hamming distance $H(\cdot, \cdot)$. There exists $\Omega' \subset \Omega$ such that $|\Omega'| \geq 2^{M/8}$ and for all distinct $\omega, \omega' \in \Omega'$, we have $H(\omega, \omega') \geq M/8$.*

**Lemma D.4** (Fano's inequality for metric estimation). *Let $\{P_\theta : \theta \in \Theta\}$ be a statistical model and let $d(\cdot, \cdot)$ be a semimetric on $\Theta$. Assume there exist $\theta_1, \ldots, \theta_M \in \Theta$ such that*

1. *(Separation) $d(\theta_i, \theta_j) \geq 2s$ for all $i \neq j$,*

2. *(Average KL control) $\frac{1}{M}\sum_{j=1}^M \mathrm{KL}(P_{\theta_j} \| P_{\theta_1}) \leq \alpha \log M$ for some $\alpha \in (0, 1/8)$.*

*Then for any estimator $\hat\theta$,*

$$\sup_{\theta \in \Theta} \mathbb{E}_\theta\left[d(\hat\theta, \theta)\right] \geq s\left(1 - \frac{\alpha \log M + \log 2}{\log M}\right) \geq \frac{s}{2} \tag{26}$$

*whenever $\log M \geq 4 \log 2$.*

*Proof.* Let $\theta$ be uniformly distributed over $\{\theta_1, \ldots, \theta_M\}$ and let $I$ be the random index. For any estimator $\hat\theta$, define the induced decoder $\hat{I} := \arg\min_{j \in [M]} d(\hat\theta, \theta_j)$ (break ties arbitrarily). By the separation assumption, the event $\{\hat{I} \neq I\}$ implies $d(\hat\theta, \theta) \geq s$. Thus,

$$\mathbb{E}[d(\hat\theta, \theta)] \geq s\, \mathbb{P}(\hat{I} \neq I).$$

By the standard Fano bound for multi-hypothesis testing (see, e.g., Tsybakov, 2009, Theorem 2.5),

$$\mathbb{P}(\hat{I} \neq I) \geq 1 - \frac{I(\theta; \mathcal{D}) + \log 2}{\log M},$$

where $\mathcal{D}$ denotes the data. Using the mutual information bound $I(\theta; \mathcal{D}) \leq \frac{1}{M}\sum_{j=1}^M \mathrm{KL}(P_{\theta_j} \| P_{\theta_1})$ and the KL condition gives (26). $\qquad\square$

### D.1.3. A CONCENTRATION LEMMA FOR VALIDATION-BASED SELECTION

The proof of Theorem 5.7 uses sample splitting: the candidate predictors are trained on $\mathcal{D}_{\mathrm{tr}}$ and evaluated on an independent validation set $\mathcal{D}_{\mathrm{val}}$. Conditional on $\mathcal{D}_{\mathrm{tr}}$, the candidates are fixed functions, so we can apply concentration on the validation sample.

**Lemma D.5** (From validation ERM to oracle inequality: a two-model version). *Assume $\epsilon$ is sub-Gaussian with parameter $\sigma$ (Gaussian is a special case) and $f^*$ is bounded: $\|f^*\|_\infty \leq M_*$. Let $\hat{f}_1, \hat{f}_2$ be two (possibly random) predictors measurable w.r.t. $\mathcal{D}_{\mathrm{tr}}$. Let $\mathcal{D}_{\mathrm{val}} = \{(X_i, Y_i)\}_{i=1}^{n_{\mathrm{val}}}$ be independent of $\mathcal{D}_{\mathrm{tr}}$, and define the validation risks*

$$\hat{L}(\hat{f}_k) := \frac{1}{n_{\mathrm{val}}}\sum_{i=1}^{n_{\mathrm{val}}} (Y_i - \hat{f}_k(X_i))^2, \qquad L(\hat{f}_k) := \mathbb{E}\left[(Y - \hat{f}_k(X))^2 \mid \mathcal{D}_{\mathrm{tr}}\right].$$

*Let $\hat{f}_{\mathrm{sel}} \in \{\hat{f}_1, \hat{f}_2\}$ minimize $\hat{L}(\cdot)$. This two-model selector is denoted $\hat{f}_{\mathrm{safe}}$ in the main text. Assume additionally that, almost surely, $\|\hat{f}_1\|_\infty \le M$ and $\|\hat{f}_2\|_\infty \le M$ for some $M < \infty$. Then there exists a constant $C > 0$ depending only on $(M, M_*, \sigma)$ such that for all $\delta' \in (0, 1)$, with probability at least $1 - \delta'$ (over $\mathcal{D}_{\mathrm{val}}$, conditional on $\mathcal{D}_{\mathrm{tr}}$),*

$$L(\hat{f}_{\mathrm{sel}}) \le \min\{L(\hat{f}_1), L(\hat{f}_2)\} + C \frac{\log(2/\delta')}{n_{\mathrm{val}}}. \tag{27}$$

*Proof.* Fix $\mathcal{D}_{\mathrm{tr}}$ and abbreviate $f_k := \hat{f}_k$ as deterministic functions. Let $\ell_k(x, y) := (y - f_k(x))^2$. Define the centered loss difference

$$\xi_i := \big(\ell_1(X_i, Y_i) - \ell_2(X_i, Y_i)\big) - \mathbb{E}\big[\ell_1(X, Y) - \ell_2(X, Y)\big],$$

where $(X, Y)$ is an independent copy from the validation distribution (conditional on $\mathcal{D}_{\mathrm{tr}}$). By the selection rule, $\hat{L}(f_{\mathrm{sel}}) \le \min\{\hat{L}(f_1), \hat{L}(f_2)\}$, hence

$$L(f_{\mathrm{sel}}) - \min\{L(f_1), L(f_2)\} \le \max_{k \in \{1,2\}} \big(L(f_k) - \hat{L}(f_k)\big) + \max_{k \in \{1,2\}} \big(\hat{L}(f_k) - L(f_k)\big) = 2 \max_{k \in \{1,2\}} |\hat{L}(f_k) - L(f_k)|.$$

It remains to control $|\hat{L}(f_k) - L(f_k)|$ for bounded predictors under sub-Gaussian noise. Write $Y = f^*(X) + \epsilon$ with $\epsilon$ sub-Gaussian and $\|f^*\|_\infty \le M_*$. Then $Y$ is sub-Gaussian up to an additive bounded shift; moreover, for fixed bounded $f_k$, the random variable $(Y - f_k(X))^2$ is sub-exponential with parameters depending only on $(M, M_*, \sigma)$. A standard Bernstein inequality for sub-exponential variables implies that for each $k$,

$$\mathbb{P}\Big(|\hat{L}(f_k) - L(f_k)| \ge t \,\Big|\, \mathcal{D}_{\mathrm{tr}}\Big) \le 2 \exp\Big(-c\, n_{\mathrm{val}} \min\{t^2, t\}\Big)$$

for a constant $c = c(M, M_*, \sigma) > 0$. Applying a union bound over $k \in \{1, 2\}$ and choosing $t \asymp \log(2/\delta')/n_{\mathrm{val}}$ yields (27). $\square$

*Remark* D.6 (About boundedness of candidate predictors). Lemma D.5 assumes $\|\hat{f}_k\|_\infty \le M$. In our setting, $\|f_0\|_\infty \le B_0$ and $\|r^*\|_\infty \le B$ imply $\|f^*\|_\infty \le B_0 + B$. For theoretical guarantees, one may replace any learned residual predictor $\hat{r}$ by its clipped version $\mathrm{clip}(\hat{r}, [-B, B])$ and correspondingly clip $f_0 + \hat{r}$ into $[-(B_0 + B), B_0 + B]$. This clipping does not increase the squared loss conditional on $X$ when $Y$ is generated from a bounded signal plus noise, and it ensures the boundedness condition required for the concentration step.

### D.2. Proof of Theorem 5.4 (Minimax Lower Bound)

We prove the lower bound via a packing construction and Fano's method, specialized to the localized class $\mathcal{F}(\delta)$ (Definition 5.3).

STEP 0: REDUCTION TO RESIDUAL REGRESSION AND CENTERING AT THE ORIGIN

By Lemma B.2 (Appendix B), estimating $f^*$ is equivalent to estimating $r^* = f^* - f_0$ from residual observations $Z_i := Y_i - f_0(X_i) = r^*(X_i) + \epsilon_i$. Moreover, because $f_0$ is fixed and known, translating the model by $f_0$ does not change the noise distribution or the design. Therefore it suffices to lower bound the minimax risk over the residual class

$$\mathcal{R}_n(\delta) := \inf_{\hat{r}} \sup_{r \in \mathcal{H}(\beta, L): \|r\|_{L_2(P_X)} \le \delta, \|r\|_\infty \le B} \mathbb{E}\big[\|\hat{r} - r\|_{L_2(P_X)}^2\big]. \tag{28}$$

We will show $\mathcal{R}_n(\delta) \gtrsim \min\{\delta^2, n^{-2\beta/(2\beta+d)}\}$, which implies the theorem.

STEP 1: A LOCALIZED PACKING VIA DISJOINT BUMPS

Fix an integer $m \ge 2$ to be chosen later and partition $[0, 1]^d$ into $M := m^d$ disjoint cubes $\{R_j\}_{j=1}^M$ of side length $1/m$. Let $x_j$ be the center of $R_j$.

Choose a fixed bump $K : \mathbb{R}^d \to \mathbb{R}$ satisfying: (i) $\mathrm{supp}(K) \subset [-1/2, 1/2]^d$, (ii) $K(0) > 0$, (iii) $K \in \mathcal{H}(\beta, 1)$ as a function on $\mathbb{R}^d$ (restricted to its support), and (iv) $\|K\|_{L_2}^2 = \int_{\mathbb{R}^d} K(u)^2 \, du \in (0, \infty)$. (Such functions exist; for example, one may take a smooth compactly supported bump and rescale its Hölder norm.)

For amplitude $h > 0$, define the bumps

$$\psi_j(x) := h\, K\big(m(x - x_j)\big), \qquad j = 1, \ldots, M, \tag{29}$$

and for $\omega \in \{0,1\}^M$ define the candidate residuals

$$r_\omega(x) := \sum_{j=1}^M \omega_j \psi_j(x). \tag{30}$$

By construction, $\operatorname{supp}(\psi_j) \subset R_j$, so the supports are disjoint across $j$.

STEP 2: VERIFY CLASS CONSTRAINTS

**(a) Hölder constraint.** Scaling properties of Hölder norms imply that $\psi_j \in \mathcal{H}(\beta, C_K h m^\beta)$ for a constant $C_K$ depending only on $K, \beta, d$. Consequently, using disjoint supports in (30) and the triangle inequality for derivatives on each cube, there exists $C_1 = C_1(K, \beta, d)$ such that

$$r_\omega \in \mathcal{H}(\beta, L) \quad \text{provided that} \quad h \leq \frac{L}{C_1}\, m^{-\beta}. \tag{31}$$

**(b) $L_2(P_X)$ constraint.** By disjoint supports and Lemma B.1 (Appendix B),

$$\|r_\omega\|_{L_2(P_X)}^2 = \sum_{j=1}^M \omega_j \|\psi_j\|_{L_2(P_X)}^2 \leq \sum_{j=1}^M \|\psi_j\|_{L_2(P_X)}^2 \leq C \sum_{j=1}^M \|\psi_j\|_{L_2}^2.$$

A change of variables $u = m(x - x_j)$ gives $\|\psi_j\|_{L_2}^2 = h^2 m^{-d} \|K\|_{L_2}^2$, hence

$$\|r_\omega\|_{L_2(P_X)}^2 \leq C\, M\, h^2 m^{-d} \|K\|_{L_2}^2 = C\, h^2 \|K\|_{L_2}^2.$$

Therefore, there exists $C_2 = C_2(K, c, C)$ such that

$$\|r_\omega\|_{L_2(P_X)} \leq \delta \quad \text{provided that} \quad h \leq \frac{\delta}{C_2}. \tag{32}$$

**(c) $\|\cdot\|_\infty$ constraint.** Since $\|\psi_j\|_\infty \leq h\|K\|_\infty$, we have $\|r_\omega\|_\infty \leq h\|K\|_\infty$. Thus $h \leq B/\|K\|_\infty$ ensures $\|r_\omega\|_\infty \leq B$.

**Choice of amplitude.** Combine (31) and (32) (and the $\|\cdot\|_\infty$ bound) by taking

$$h := \kappa \cdot \min\big\{Lm^{-\beta}, \delta\big\}, \tag{33}$$

for a sufficiently small constant $\kappa = \kappa(K, \beta, d, c, C, B) \in (0, 1)$.

STEP 3: SEPARATION VIA VARSHAMOV–GILBERT

By Lemma D.3, there exists $\Omega \subset \{0,1\}^M$ with $|\Omega| \geq 2^{M/8}$ such that for all distinct $\omega, \omega' \in \Omega$, $H(\omega, \omega') \geq M/8$. Using disjoint supports and Lemma B.1,

$$\|r_\omega - r_{\omega'}\|_{L_2(P_X)}^2 = \sum_{j=1}^M (\omega_j - \omega_j')^2 \|\psi_j\|_{L_2(P_X)}^2 \geq \frac{M}{8} \min_j \|\psi_j\|_{L_2(P_X)}^2 \geq \frac{M}{8} c \|\psi_1\|_{L_2}^2.$$

Since $\|\psi_1\|_{L_2}^2 = h^2 m^{-d} \|K\|_{L_2}^2$ and $M = m^d$, we conclude that

$$\|r_\omega - r_{\omega'}\|_{L_2(P_X)}^2 \geq c_0\, h^2 \tag{34}$$

for some constant $c_0 = c_0(K, c) > 0$.

STEP 4: KL CONTROL

For $\omega, \omega' \in \Omega$, by Lemma D.2,

$$\mathrm{KL}(P_{r_\omega} \| P_{r_{\omega'}}) = \frac{n}{2\sigma^2} \| r_\omega - r_{\omega'} \|_{L_2(P_X)}^2 \leq \frac{n}{2\sigma^2} \max_{\omega \neq \omega'} \| r_\omega - r_{\omega'} \|_{L_2(P_X)}^2 \leq C_3 \frac{nh^2}{\sigma^2}, \tag{35}$$

where $C_3$ is a constant depending only on $(K, C)$.

STEP 5: APPLY FANO AND OPTIMIZE OVER $m$

Let $d(\cdot, \cdot)$ be the semimetric $d(r, r') := \| r - r' \|_{L_2(P_X)}$. By (34), the packing is separated by $2s$ with $s := \frac{1}{2} \sqrt{c_0} \, h$. Moreover, choosing $m$ so that the average KL satisfies the Fano condition in Lemma D.4, namely

$$C_3 \frac{nh^2}{\sigma^2} \leq \alpha \log |\Omega| \quad \text{with} \quad \log |\Omega| \geq \frac{M}{8} \log 2, \tag{36}$$

implies (for a sufficiently small absolute $\alpha$) that

$$\inf_{\hat{r}} \sup_{r \in \{r_\omega : \omega \in \Omega\}} \mathbb{E} \| \hat{r} - r \|_{L_2(P_X)} \gtrsim h, \quad \text{hence} \quad \inf_{\hat{r}} \sup_{r \in \{r_\omega : \omega \in \Omega\}} \mathbb{E} \| \hat{r} - r \|_{L_2(P_X)}^2 \gtrsim h^2.$$

It remains to choose $m$ to maximize $h^2$ subject to (33) and (36).

**Case 1: sample-dominated regime.** Suppose $\delta \geq c \, n^{-\beta/(2\beta+d)}$ (for a small constant $c$ depending on fixed parameters). Choose $m \asymp n^{1/(2\beta+d)}$ and $h \asymp Lm^{-\beta} \asymp n^{-\beta/(2\beta+d)}$. Then $M = m^d \asymp n^{d/(2\beta+d)}$ and $nh^2 \asymp n^{d/(2\beta+d)} \asymp M$, so (36) holds for a suitable choice of constants. Thus the minimax risk is lower bounded by $h^2 \asymp n^{-2\beta/(2\beta+d)}$.

**Case 2: black-box-dominated regime.** Suppose $\delta \leq c \, n^{-\beta/(2\beta+d)}$. Set $h \asymp \delta$ (consistent with (33)), and choose $m$ to satisfy the KL/Fano feasibility: from (35) and $\log |\Omega| \asymp M = m^d$, it suffices to take $m^d \gtrsim n\delta^2$, i.e. $m \gtrsim (n\delta^2)^{1/d}$. On the other hand, the Hölder constraint requires $h \lesssim Lm^{-\beta}$, i.e. $m \lesssim (L/\delta)^{1/\beta}$. Feasibility of both bounds is equivalent (up to constants) to $(n\delta^2)^{1/d} \lesssim (L/\delta)^{1/\beta}$, which rearranges to $\delta \lesssim n^{-\beta/(2\beta+d)}$ (absorbing fixed constants into $c$), exactly the present case. Therefore we can pick such an $m$ and conclude the risk lower bound is $h^2 \asymp \delta^2$.

Combining the two cases yields

$$\mathcal{R}_n(\delta) \gtrsim \min \left\{ \delta^2, \ n^{-2\beta/(2\beta+d)} \right\},$$

which proves Theorem 5.4. $\qquad \square$

## D.3. Proof of Theorem 5.5 (Upper Bound)

We construct an estimator that adaptively achieves the minimum of the black-box error floor $\delta^2$ and the optimal nonparametric rate for learning the residual.

STEP 1: CANDIDATE ESTIMATORS

Split the labeled sample into $\mathcal{D}_{\mathrm{tr}}$ and $\mathcal{D}_{\mathrm{val}}$ with $n_{\mathrm{val}} \asymp n$. Consider two candidates:

$$\hat{f}_{\mathrm{BB}}(x) := f_0(x), \qquad \hat{f}_{\mathrm{Res}}(x) := f_0(x) + \hat{r}(x),$$

where $\hat{r}$ is trained on $\mathcal{D}_{\mathrm{tr}}$ using the residual responses $Z_i = Y_i - f_0(X_i)$.

STEP 2: RISK OF THE BLACK-BOX CANDIDATE

By Definition 5.3 (and Lemma B.2), for any $f^* = f_0 + r^* \in \mathcal{F}(\delta)$,

$$\mathcal{R}(\hat{f}_{\mathrm{BB}}, f^*) = \mathbb{E} \| f_0 - f^* \|_{L_2(P_X)}^2 = \| r^* \|_{L_2(P_X)}^2 \leq \delta^2. \tag{37}$$

STEP 3: RISK OF A MINIMAX-OPTIMAL RESIDUAL REGRESSOR

It remains to upper bound the minimax risk for estimating $r^*$ in $\mathcal{H}(\beta, L)$ under random design with sub-Gaussian noise. A classical result (Stone's theorem and subsequent refinements) ensures the existence of estimators attaining the optimal rate.

**Lemma D.7** (Existence of a minimax-optimal estimator for Hölder regression). *Under Assumption 5.1, for the regression model $Z_i = r(X_i) + \epsilon_i$ with $\epsilon_i \sim \mathcal{N}(0, \sigma^2)$, there exists an estimator $\hat{r}$ (measurable w.r.t. $\mathcal{D}_{\mathrm{tr}}$) such that*

$$\sup_{r \in \mathcal{H}(\beta, L):\ \|r\|_\infty \leq B} \mathbb{E}\big[\|\hat{r} - r\|^2_{L_2(P_X)}\big] \leq C_{up}\, n_{\mathrm{tr}}^{-\frac{2\beta}{2\beta+d}}, \tag{38}$$

*where $C_{up}$ depends only on $(\beta, L, d, \sigma, c, C, B)$.*

*Proof sketch with precise dependency.* This is a standard nonparametric regression upper bound for Hölder smoothness under random design with density bounded away from $0$ and $\infty$. One may take, for example, a local polynomial estimator of degree $\lfloor \beta \rfloor$ or an appropriately chosen series/wavelet estimator, which attains the minimax rate $n^{-2\beta/(2\beta+d)}$ in $L_2(P_X)$ risk; see, e.g., (Tsybakov, 2009, Chapter 2) and the classical optimality result of (Stone, 1982). The constants depend on $(\beta, L, d, \sigma)$ and on the density bounds $(c, C)$ through norm equivalences (Lemma B.1). $\square$

Applying Lemma D.7 to $r^*$ yields

$$\sup_{f^* \in \mathcal{F}(\delta)} \mathcal{R}(\hat{f}_{\mathrm{Res}}, f^*) = \sup_{r^*} \mathbb{E}\|\hat{r} - r^*\|^2_{L_2(P_X)} \leq C_{up}\, n_{\mathrm{tr}}^{-\frac{2\beta}{2\beta+d}}. \tag{39}$$

STEP 4: VALIDATION SELECTION AND CONCLUSION

Define $\hat{f}_{\mathrm{safe}}$ as the validation-selected predictor between $\hat{f}_{\mathrm{BB}}$ and $\hat{f}_{\mathrm{Res}}$, as in Algorithm 1. By Lemma D.5 (applied conditionally on $\mathcal{D}_{\mathrm{tr}}$), with probability at least $1 - \delta'$ over $\mathcal{D}_{\mathrm{val}}$,

$$L(\hat{f}_{\mathrm{safe}}) \leq \min\{L(\hat{f}_{\mathrm{BB}}), L(\hat{f}_{\mathrm{Res}})\} + C\frac{\log(2/\delta')}{n_{\mathrm{val}}},$$

for $C$ depending only on $(B_0, B, \sigma)$ (via boundedness and sub-Gaussianity). Taking expectations and using (37) and (39) gives

$$\sup_{f^* \in \mathcal{F}(\delta)} \mathcal{R}(\hat{f}_{\mathrm{safe}}, f^*) \leq C' \min\left\{\delta^2,\ n_{\mathrm{tr}}^{-\frac{2\beta}{2\beta+d}}\right\} + C''\frac{1}{n_{\mathrm{val}}},$$

where we fixed $\delta'$ as a constant (e.g., $\delta' = 1/4$) to absorb $\log(2/\delta')$ into $C''$, and used $n_{\mathrm{tr}} \asymp n$ and $n_{\mathrm{val}} \asymp n$. This yields the claimed bound in Theorem 5.5 (up to constant-factor changes). $\square$

**D.4. Proof of Corollary 5.6**

The proof follows directly from the risk decomposition and the sample splitting construction. Algorithm 1 constructs two candidates: $\hat{f}_{\mathrm{BB}} = f_0$ and $\hat{f}_{\mathrm{Res}} = f_0 + \hat{r}$, where $\hat{r}$ is trained on $\mathcal{D}_{\mathrm{tr}}$ with sample size $n_{\mathrm{tr}} = (1 - \rho)n$.

From Theorem 5.5 (specifically the oracle inequality argument in its proof), the risk of the validation-selected estimator $\hat{f}_{\mathrm{safe}}$ satisfies:

$$\sup_{f^* \in \mathcal{F}(\delta)} \mathbb{E}\|\hat{f}_{\mathrm{safe}} - f^*\|^2_{L_2(P_X)} \leq C \cdot \min\left\{\delta^2, n_{\mathrm{tr}}^{-\frac{2\beta}{2\beta+d}}\right\} + \frac{C'}{n_{\mathrm{val}}}. \tag{40}$$

Substituting $n_{\mathrm{tr}} = (1 - \rho)n$ into the estimation error term:

$$n_{\mathrm{tr}}^{-\frac{2\beta}{2\beta+d}} = ((1-\rho)n)^{-\frac{2\beta}{2\beta+d}} = (1-\rho)^{-\frac{2\beta}{2\beta+d}} \cdot n^{-\frac{2\beta}{2\beta+d}}. \tag{41}$$

The additive selection cost term is $C'/n_{\mathrm{val}} = C'/(\rho n) = O(n^{-1})$. Since the nonparametric rate $n^{-\frac{2\beta}{2\beta+d}}$ is asymptotically slower than $n^{-1}$ (for any finite $d \geq 1$), the $O(n^{-1})$ term is lower order, although it remains an explicit finite-sample validation-selection cost.

Thus, we obtain:

$$\sup_{f^* \in \mathcal{F}(\delta)} \mathbb{E}\|\hat{f}_{\text{safe}} - f^*\|^2 \leq (1-\rho)^{-\frac{2\beta}{2\beta+d}} \cdot C \min\left\{\delta^2, n^{-\frac{2\beta}{2\beta+d}}\right\} + \tilde{O}(n^{-1}). \tag{42}$$

For the specific case of $\rho = 0.2$ and $\beta \approx d$ (e.g., $d = 1, \beta = 1$), the exponent is $-2/3$. The inflation factor is $(0.8)^{-2/3} \approx 1.16$. Generally, for $\beta, d > 0$, this factor is a constant independent of $n$. This confirms that Algorithm 1 is minimax-optimal up to the validation-selection cost and a constant factor from sample splitting. □

### D.5. Proof of Theorem 5.7 (Oracle Inequality for Safe Selection)

Theorem 5.7 is a direct application of Lemma D.5 with $\hat{f}_1 = \hat{f}_{\text{BB}} = f_0$ and $\hat{f}_2 = \hat{f}_{\text{Res}} = f_0 + \hat{r}$. Indeed, conditional on $\mathcal{D}_{\text{tr}}$, both candidates are fixed functions, and the validation set is independent. The boundedness assumptions needed for Lemma D.5 are satisfied by $\|f_0\|_\infty \leq B_0$ and by clipping $f_0 + \hat{r}$ into $[-(B_0+B), B_0+B]$ if necessary (Remark following Lemma D.5).

Therefore, for any $\delta' \in (0, 1)$, with probability at least $1 - \delta'$,

$$\mathcal{R}(\hat{f}_{\text{safe}}) \leq (1+\gamma)\min\{\mathcal{R}(\hat{f}_{\text{BB}}), \mathcal{R}(\hat{f}_{\text{Res}})\} + C\frac{\log(2/\delta')}{n_{\text{val}}},$$

for constants $(\gamma, C)$ depending only on boundedness and the sub-Gaussian noise parameter. This is exactly the statement of Theorem 5.7 (up to renaming constants). □

### D.6. The Price of Safety: Sample Splitting Cost

Here we justify the inflation factor discussed in Section 5.5. In the sample-dominated regime, the residual estimation error is of order $n_{\text{tr}}^{-2\beta/(2\beta+d)}$. With $n_{\text{tr}} = (1-\rho)n$, we have

$$n_{\text{tr}}^{-\frac{2\beta}{2\beta+d}} = \big((1-\rho)n\big)^{-\frac{2\beta}{2\beta+d}} = (1-\rho)^{-\frac{2\beta}{2\beta+d}} n^{-\frac{2\beta}{2\beta+d}}.$$

Thus the sole effect of sample splitting on the main nonparametric term is a multiplicative constant $(1-\rho)^{-2\beta/(2\beta+d)}$. In addition, the oracle inequality contributes an additive selection penalty of order $O(1/n_{\text{val}}) = O(1/(\rho n))$, which is asymptotically negligible compared to $n^{-2\beta/(2\beta+d)}$ whenever $2\beta/(2\beta+d) < 1$ (i.e., for any finite $d$ and $\beta > 0$).

## E. Theoretical Novelty and Comparison to Classical Localization

### E.1. Comparison to Classical Localized Minimax Analysis

Standard non-parametric regression typically considers the minimax risk over a global Hölder ball $\mathcal{H}(\beta, L)$. The classical minimax rate is known to be $n^{-\frac{2\beta}{2\beta+d}}$ (Stone, 1982; Tsybakov, 2009).

Our work introduces a novel twist by considering the intersection of the Hölder ball with an $L_2$-neighborhood of a fixed predictor $f_0$:

$$\mathcal{F}(\delta) = \mathcal{H}(\beta, L) \cap \{f : \|f - f_0\|_{L_2} \leq \delta\}.$$

This formulation fundamentally alters the metric entropy of the hypothesis class.

- **Classical Regime:** When $\delta$ is large (specifically $\delta > n^{-\frac{\beta}{2\beta+d}}$), the $L_2$ constraint is inactive. The covering number $\log N(\epsilon, \mathcal{F}(\delta))$ behaves like $\epsilon^{-d/\beta}$, recovering the standard rate.

- **Black-Box Regime:** When $\delta$ is small, the geometry of $\mathcal{F}(\delta)$ is dominated by the $L_2$ ball. The local entropy integral is truncated, leading to a parametric-like rate limited by $\delta^2$.

While localized analysis (e.g., local Rademacher complexity) deals with adaptivity to the *target function's* local smoothness, our analysis characterizes adaptivity to the *prior's* quality. The phase transition at $\delta_c(n)$ is the precise mathematical boundary where the "information content" of the samples $n$ supersedes the "information content" of the prior constraint $\delta$.

# F. Implementation Details for Reproducibility

To ensure full reproducibility, we provide the specific hyperparameters and architectures used in our experiments.

## 1. Synthetic Experiments (KRR)

- **Model:** Kernel Ridge Regression with RBF kernel $k(x, x') = \exp(-\gamma \|x - x'\|^2)$.

- **Hyperparameters:**
    - Regularization $\alpha$: Grid search over $\{10^{-4}, 10^{-3}, 10^{-2}, 10^{-1}\}$.
    - Kernel Gamma $\gamma$: Grid search over $\{1, 5, 10, 20\}$.

- **Selection:** 5-fold cross-validation on the training split.

## 2. Vision Experiments (ResNet/MLP)

- **Backbone:** CLIP (ViT-B/32) frozen image encoder.

- **Residual Head Architecture:**
    - Input: 512-dim CLIP features.
    - Hidden: Linear(512, 512) $\rightarrow$ ReLU $\rightarrow$ Dropout(0.3).
    - Output: Linear(512, NumClasses).

- **Optimization:** AdamW optimizer, Learning rate $10^{-3}$, Weight decay $10^{-4}$.

- **Training:** Batch size 32 (for $n < 1000$) or 64. Early stopping with patience=10 on validation accuracy.

- **Zero-Initialization:** The weights and biases of the final output layer are explicitly set to 0.0 at initialization.

## 3. NLP Experiments (Qwen)

- **Backbone:** Qwen3-8B (frozen).

- **Feature Extraction:** Last token hidden state from the last layer.

- **Head Architecture:** Same MLP structure as Vision.

- **Prompt for $f_0$:** "Classify the article into one of: {labels}. Article: {text} Category:"

### F.1. Computational Complexity Analysis

A potential concern with two-stage methods is computational overhead. We clarify that Algorithm 1 incurs negligible cost compared to standard baselines:

- **Training Cost:** The residual model $\hat{r}$ has the same architecture and training complexity as learning from scratch. The black-box $f_0$ is frozen and only requires a one-time forward pass to cache features/logits.

- **Selection Cost:** The safe selection step requires only one forward pass on the validation set to compute two scalars: $\mathcal{L}_{\mathrm{BB}}$ and $\mathcal{L}_{\mathrm{Res}}$. This is $O(n_{\mathrm{val}})$.

- **Comparison with Weighted Ensemble:** The WGT(Val-Tuned) baseline requires a grid search over $\alpha \in [0, 1]$ to minimize validation error. Our method requires no hyperparameter search for the combination mechanism, making it computationally cheaper and more stable than the ensemble baseline.

# G. Ablation Studies

We conduct additional ablation studies to verify the sensitivity of our method to the validation split ratio $\rho$ and the impact of zero-initialization.

### G.1. Sensitivity to Validation Fraction $\rho$

We investigate the impact of the validation split ratio $\rho = n_{\mathrm{val}}/n$ on the final performance using the 1D synthetic dataset ($n = 200, \delta = 0.2$). As shown in Figure 3a, the risk is stable for $\rho \in [0.15, 0.3]$.

- When $\rho$ is too small ($< 0.1$), the validation set is insufficient to reliably estimate the risk, leading to noisy selection.

- When $\rho$ is too large ($> 0.5$), the training set becomes too small to learn the residual effectively.

Our choice of $\rho = 0.2$ in the main experiments strikes an optimal balance.

### G.2. Effect of Zero-Initialization

We compare our zero-initialization strategy against standard random initialization (Kaiming Uniform) on CIFAR-100 ($n = 500$). Figure 3b tracks the $L_2$ norm of the residual head's weights during training.

- **Random Init:** The residual norm starts large, causing the initial predictor $\hat{f} = f_0 + \hat{r}$ to deviate significantly from the black-box $f_0$. This high initial variance increases the risk of early overfitting and negative transfer.

- **Zero Init (Ours):** The norm starts at 0 and grows gradually. This acts as a strong inductive bias, ensuring the model starts exactly at the black-box prior and only deviates when the data provides sufficient evidence. This leads to a lower final test error (Accuracy: **61.0%** vs. 58.2% for Random Init).

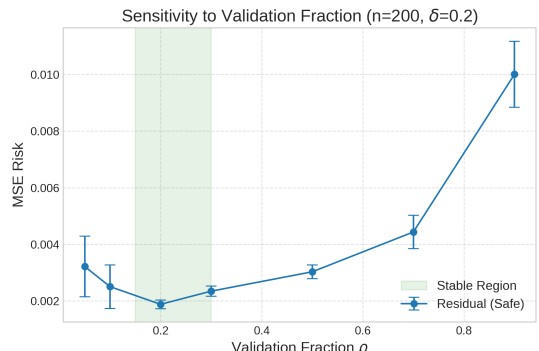
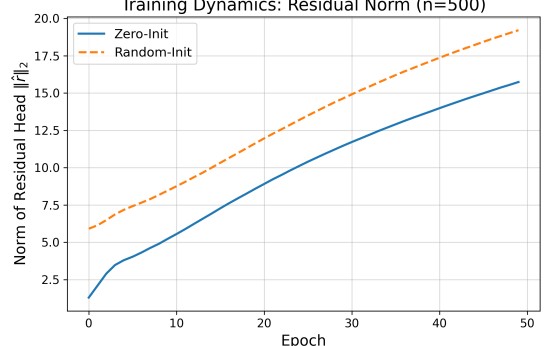

*(a)* Validation fraction $\rho$ and MSE risk.    *(b)* Residual head norm $\|\hat{r}\|_2$ during training.

*Figure 3.* Ablation studies. Left: the validation split is stable for $\rho \in [0.15, 0.3]$. Right: zero-initialization ensures a smooth departure from the prior $f_0$.

