# OpenReview forum: "Black-Box Assisted Regression: Phase Transitions and Minimax Optimality"
_ICML.cc/2026/Conference — ICML 2026 regular_

### Official Review · Reviewer_H8jD · 2026-02-16

**Soundness:** 2
**Presentation:** 1
**Significance:** 3
**Originality:** 2
**Overall Recommendation:** 4
**Confidence:** 3

**Summary:**

This paper provides a theoretical iteration of some of the (possibly known concepts) in transfer learning.
The authors assume a problem formulation that is learning a residual $\hat{r}$ in estimating a presumed true function $f^* = f_0 + \hat{r}$.
The work progresses by providing Algorithm 1 called "Safe Black-Box Residual Estimator" which uses a constant $\hat{\alpha}$ that is set to either 0 or 1 according to the comparative performance of $f_0$ and the residual estimator $f_0 + \hat{r}$ on $\mathcal{D}_val$ which is trained on $\mathcal{D}_tr$.
They provide interesting theoretical analysis of the algorithm under such setup. They provide a experimental evaluation and visualization of their claims on synthetic, vision and language datasets.

**Compliance With Llm Reviewing Policy:**

Affirmed.

**Final Justification:**

I would like to thank authors for their time.

I had a low rating for this paper originally. As acknowledged by authors in the rebuttal, it was due to the blurry scope of the paper of which I tried to give a detailed review. The authors have promised a sharper abstract and introduction in the rebuttal and their last comment. I think this work is a good contribution after the promised changes materialized, consequently, I increased my earlier rating.

**Key Questions For Authors:**

1. I would like to see how this method compares to existing baselines. Adding four to five recent baselines would improve the chances of higher rating from my side.
2. In L268 we have "mitigating negative transfer". Well, the word "mitigating" is weaker than NO negative transfer. I would like to know why we do not see any degradation although $\delta$ is unknown (for example) Fig 2? It is logical to expect that there might errors in finding $\hat{c}$ (as it is random), so not seeing any degradation in Fig 2 Tab 3 Tab 4 is unexpected. Are the authors selecting the best performance across many runs? if that is the case the theoretical analysis and algorithm and presentation should be updated to reflect that.
3. In Alg 1 the alpha is boolean. This means that the algorithm would coincide with either $f_0$ or the residual $f_0 + \hat{r}$. It seems that $\hat{\alpha}$ is only zero for n=100 (in Fig 2) and 1 otherwise. This seems to be a very simple improvement on a pretrained classifier. I'd like to ask the authors to discuss the novelty of their idea in more detail.

**Limitations:**

Under the discussion section 7, the authors have included one limitation of their work which is assuming homoscedastic noise.

**Strengths And Weaknesses:**

+ **Soundness:**
    1. The authors should provide several citations for their claims. For example when claiming something is common practice or dividing the prior work into two categories in Sec. Method L092 "two extremes ...".
    2. It is hard to confirm the soundness of the improvement claims when there are not strong and recent baselines. The authors should provide baselines and compare their setup with existing work. This allows the readers to rely on the improvement claims.
    3. While the theoretical analysis is well separated into theorems, assumptions and definitions, it is better if the theoretical work is compared with the existing theoretical analysis and highlight the novelty more clearly. Furthermore, the connection between the theoretical analyses is missing and unclear. It helps readers to better understand the paper if the paper follows a narrative rather than individual theorems.
+ **Presentation:**
    The paper is generally immature in terms of presentation, to the point that it is hard to follow
1. abstract:
	1. L16 The paper starts with several unspecified variables in the abstract, this is better be an intuitive discussion about the findings and consequences.
	2. L18 "Safe Black-Box": what does safe mean here?
	3. L19 what is "zero initialization" in this context?
	4. L22 what is "negative transfer"?
	5. L24 "validate the theory": what is the theoretical result being validated?
	6. I can say that abstract is not well written.
2. introduction:
	1. I'm not sure if it is due to my pdf viewer but the citations do not seem to be correctly linked.
	2. L35 "scenario common": should be "common scenario"
	3. L35: "scientific discovery": the discussion that comes after this, does not pertain to scientific discovery.
	4. Par 1: I don't know what is the takeaway of this paragraph. the first paragraph in intro is really important and should set the stage for the rest of the text.
	5. Par2: the transition from par 1 to par 2 should be more segue.
	6. Par2: how is the problem you are trying to solve is different from transfer learning? or model distillation?
	7. right col L09: "... learning” heuristically."  this requires citation.
	8. right col L29: "prior and sample-dominated regimes" this is again unspecified.
	9. right col L30 , 32: "Safe" and "negative transfer" are undefined
	10. I can say that introduction is not well written.
3. related work:
	1. it is better to group related works by topic and distinguish your work.
	2. I am still unsure about what is the problem being solved and how it is different from prior work at the end of related work.
4. Problem Setup:
	1. L67 "with random design": this is unclear what random design is, and how it is related to assumption 5.1. From what I understand, this assumption states that the probability distribution has full support and bounded (L139).
	2. I don't see why Definition 3.1 is here. Maybe it should be moved close to where it is used.
5. Method:
	1. L92 "Standard approaches typically..." this paragraph needs more citations (at least two). One highly cited paper for each case, that is considered standard.
	2. L108 "systematic bias" is irrelevant in this context, otherwise there should be definitions and experiments showing that you can mitigate systematic bias with this approach.
	3. In the related work, after reading "typically assumes access to source data or parameters" it was unclear what your paper assumes. I had to figure that out in L57 right col "We split the available labeled data".
	4. While your definition of safety remains unclear in the paper, it seems "Safe Selection" step in the alg 1 is a marginal constant that weighs the residual classifier. It is good to compare with works on noisy label correction and estimation.
6. Implementation Details
	1. This paragraph is better to be close to experiments section as it is part of the experimental setup.
	2. L096 Zero-Initialization is really implementation details. This shows that the experimental setup and implementation details are not well separated.
	3. L100 right col "significant robustness..." this line requires citation.
	4. L103 right col: this does not have to be bold "Detailed hyperparameters ..."
	5. This subsection "Theoretical Limitation of Linear Ensembling" does not pertain to the section title.
	6. L107 "A common..." this line requires citations.
+ **Significance:** The problem of transfer learning is an interesting problem and there is plenty of room for improvement in different fronts. I asses that the paper to be about a problem with high significance.
+ **Originality:**
    1. While the theoretical analysis is helpful for understanding the Alg 1, it is a simple algorithm after all. It is effectively an interpolation between the residual and the pretrained model. I doubt that it has not been studied (at least experimentally). It is better to include citations with the prior work and distinguish the novelty in their work.
    2. Maybe the main reason for my decision (reject) is that the work does not have any baselines.

---

> ### Author Rebuttal · Authors · 2026-03-24
>
> Thank you for the detailed review and for the specific structural suggestions. We genuinely appreciate the time invested in reading through the abstract, introduction, and related work line by line—these have helped us see where the manuscript fails to communicate its intentions. We address the core points below.
>
> **On the nature of the contribution and experimental design:**
>
> We would like to clarify the nature of this work. This is a statistical learning theory paper. The main contributions are Theorems 5.4, 5.5, and 5.7—a minimax lower bound, a matching upper bound, and an oracle inequality—which together characterize a phase transition that, to our knowledge, has not appeared in prior work. Algorithm 1 is deliberately simple because its purpose is to be the object of theoretical analysis, not a competing heuristic.
>
> Accordingly, the experiments in Section 6 are numerical verifications of the theoretical predictions, following the standard template for theory papers at ICML/NeurIPS (e.g., Cai & Pu 2024, Kim et al. 2024). Each method in our experiments corresponds to a specific mathematical quantity in the theory:
>
> | Experiment | Mathematical role |
> |---|---|
> | BB-only | Realizes the prior-error floor $\delta^2$, the first term in Theorem 5.4 |
> | Scratch | Realizes the standard nonparametric rate $n^{-2\beta/(2\beta+d)}$, the second term in Theorem 5.4 |
> | Weighted ensemble | Tests the geometric limitation analyzed in Section 4.3 (Eq. 4) |
> | Residual (Safe) | Tests the upper bound of Theorem 5.5 and the oracle inequality of Theorem 5.7 |
>
> The question the experiments answer is: does the empirically observed risk track the theoretically predicted phase transition? The answer is yes (Figure 1b, Tables 1–2). This is a different goal from benchmark comparison, and adding methods from different problem settings (e.g., continual learning, weight-space adaptation) would not test our theorems. We acknowledge, however, that the manuscript does not make this distinction clearly enough, and we will restructure Section 6 to state the purpose of each experiment upfront.
>
> **On novelty:**
>
> We appreciate the reviewer's observation that Algorithm 1 is simple. We agree—and this is intentional. The novelty lies not in algorithmic complexity but in the theoretical results the algorithm enables: (i) the first minimax characterization of black-box-assisted nonparametric regression, (ii) a sharp phase transition at $\delta_c(n) \asymp n^{-\beta/(2\beta+d)}$, and (iii) a formal no-negative-transfer guarantee (Theorem 5.7). We will restructure the narrative to make this unmistakable.
>
> **On the "no degradation" concern:**
>
> We are not selecting best runs. Tables 3 and 4 report mean $\pm$ std over three independent runs. All code for reproducing every table and figure has been included in the Supplementary Material, and we invite the reviewer to verify this independently. The stability has two complementary explanations. Formally, Theorem 5.7 guarantees that the selected predictor's risk is at most $(1+\gamma)$ times the better candidate's risk plus a vanishing $O(1/n_{\mathrm{val}})$ term. Empirically, zero-initialization provides a complementary mechanism: the residual head starts at zero, so the initial predictor equals $f_0$ exactly. In few-shot regimes, gradients are insufficient to move weights far from zero within the early-stopped training budget, so the predictor stays near $f_0$ even before holdout selection applies. We will separate these two mechanisms clearly in the revision.
>
> **On boolean $\hat{\alpha}$:**
>
> The binary selection is by design: it enables the clean oracle inequality in Theorem 5.7. Continuous interpolation would require estimating a mixing weight from limited data, introducing the same instability observed in the weighted ensemble (Table 4, $n=64$). We will make clear that the algorithm's simplicity is a feature of the theoretical framework rather than a limitation.
>
> **On presentation:**
>
> We take these suggestions seriously. We will rewrite the abstract to explicitly define our three key terms: "safe" (validation-based selection that reverts to $f_0$ if held-out evidence is weak), "negative transfer" (performing worse than $f_0$ alone), and "zero-initialization" (setting residual head weights to zero so the initial predictor equals $f_0$). In the introduction, we will remove the misleading "scientific discovery" framing and clearly distinguish our fixed function-access setting from transfer learning and distillation. We will also address all formatting and structural suggestions: moving Implementation Details and Definition 3.1, retitling Section 4.3, clarifying the "random design" assumption (L67), adding missing citations, and toning down unsupported claims (L100, L108).
>
> Thank you again for the thorough and constructive feedback. The line-by-line suggestions will substantially improve the revised manuscript. We are grateful for the reviewer's engagement with this work.

---

> > ### Author Rebuttal · Reviewer_H8jD · 2026-04-01
> >
> > I would like to thank the authors for clarifications and acknowledging this reviewer's concerns. I think the authors have addressed most of my questions.
> >
> > I think most of my questions were arising from the blurry scope of the paper, partially induced by terms such as "scientific discovery" and lack of an early discussion before or after the algorithm that clarifies that the algorithm is simple by design and aimed only for verification. These will hopefully be sharper in the rewrite that authors have promised.
> >
> > I'll adjust my early rating accordingly, and thank the authors for their effort and time.

---

> > > ### Author Response · Authors · 2026-04-01
> > >
> > > You have identified exactly the root issue: the manuscript's framing obscured the theoretical nature of the contribution. Your detailed line-by-line feedback was instrumental in helping us see this clearly. In the revision, we will ensure that:
> > >
> > > 1. The abstract and introduction immediately establish this as a learning theory paper with precise definitions of all key terms.
> > > 2. The discussion surrounding Algorithm 1 explicitly states that its simplicity is by design—it serves as the object of theoretical analysis, not a proposed heuristic.
> > > 3. Section 6 opens with a clear statement that the experiments are numerical verifications of the theorems, with each method's mathematical role stated upfront.
> > >
> > > Your review has materially improved the paper. We are grateful for both the initial scrutiny and the willingness to engage with our response.

---

### Official Review · Reviewer_GSQs · 2026-03-10

**Soundness:** 3
**Presentation:** 2
**Significance:** 3
**Originality:** 2
**Overall Recommendation:** 5
**Confidence:** 3

**Summary:**

This paper proposes a regression method that incorporates a given black-box prior function $f_0$ without suffering from negative transfer.
The proposed approach performs non-parametric regression on the residual $y_i - f_0(x_i)$ to refine the prediction of $f_0$, but only accepts this solution when its empirical performance is superior to that of $f_0$ on a hold-out dataset (Algorithm 1).

The authors motivate this residual learning approach by explaining its theoretical advantage over taking the convex combination of $f_0$ and an estimate $\hat{f}_{\text{scratch}}$  obtained by regression performed directly on $y_i$.

The paper then theoretically studies this residual regression problem by giving a minimax lower bound (Theorem 5.4) under some regularity conditions including the one of Holder class. The rate of this bound is the minimum between $\delta^2$ (the square distance of $f_0$ to the true function) and the rate $n^{-\frac{2\beta}{2\beta + d}}$ that commonly appears in the standard non-parametric regression in a similar setup.

To have an estimator matching the lower bound, it should be adaptive to the two cases: $\delta$ is small enough for $f_0$ to be reliable, or it is too large so that it is better to rely on the training data. The proposed method does this by switching between the two modes by comparing their validation errors. In deed, Theorem 5.5 shows that an algorithm similar to Algorithm 1 matches the minimax lower bound up to the extra $O(1/n)$ term.

Theorem 5.7 illustrates the fact that the proposed method leverages the access to $f_0$ without negative transfer by achieving the minimum between the two errors, up to the $O(1/n)$ term. The authors explain that sample splitting for selecting the better estimate only incurs a constant factor of increase, not far from 1 when the regularity matches the dimension.

Finally, the paper presents numerical experiments including image classification and an NLP task, demonstrating the effectiveness of the proposed method.

**Compliance With Llm Reviewing Policy:**

Affirmed.

**Final Justification:**

I had a few questions for clarification, which have been addressed by the authors' rebuttal. I updated the overall recommendation score accordingly: 4 --> 5.
The methodological idea is rather simple, but the paper presents interesting theoretical findings and great empirical evaluation.

If the paper is accepted, I hope the authors include the writing improvements that they suggested during the rebuttal.

**Key Questions For Authors:**

- I would like to hear some comments about the first two points in the Weaknesses section (if there are some confusion on my side).

- The authors explain the discrepancy of the types of the loss functions used in the theory part and the experiment part as follows.
> our theoretical insights about phase transitions and the advantages of residual learning remain applicable

Does this mean you have complete proofs for this version of algorithm for the corresponding theorems?

- In Figure 2, how does the proposed method go beyond both Zero-shot and Scratch, if it only switches between these two?

**Limitations:**

yes

**Strengths And Weaknesses:**

### Strengths
- This work is motivated by modern practical scenarios where there is a pretrained model available and one wants to fine-tune it for the specific task.

- The paper provides solid theoretical guarantees for the proposed method.

- The work also studies the minimax lower bound, which is useful to understand the performance of the proposed method relative to the hardness of the problem.

- The proposed method is proven to avoid the negative transfer, which is well demonstrated both theoretically and empirically.

- The experiments include practical scenarios of computer vision and NLP using pretrained models.

### Weaknesses
- From the presentation, it is not clear whether the theoretical results are asymptotic ones or non-asymptotic ones. If I understand the setup well, $f_0$ is pre-trained on another dataset with some bias. So, $\delta$ is not a function of $n$ and does not tend to zero. In this case, $\text{min}\\{\delta^2, n^{-2\beta/(2\beta + d)}\\}$ is asymptotically equivalent to $n^{-2\beta/(2\beta + d)}$. It seems to me that the results are only meaningful if they are non-asymptotic.

- The difference between the estimators considered in Algorithm 1, Theorem 5.5, and Theorem 5.7 are not clear to me from the paper.

- The paper claims that the proposed method achieves the minimax optimality, but the upper bound and the minimax lower bound do not exactly match each other. The most important terms do match each other, but the claim should be stated in a more precise way.

- The paper seems to use different ways (or notation) of referring to the proposed method or the proposed estimate, which is a little confusing.

---

> ### Author Rebuttal · Authors · 2026-03-24
>
> Thank you for the careful review and for engaging closely with the theoretical claims. We are glad that the main idea—safe residual correction around a black-box prior—came across, and we agree that several points should be clarified more explicitly in the paper.
>
> **On asymptotic vs. non-asymptotic (Weakness 1):**
> > *$\delta$ is not a function of $n$ and does not tend to zero... results are only meaningful if they are non-asymptotic.*
>
> The main results are intended as non-asymptotic risk statements in $n$, $\delta$, $\beta$, and $d$. You are absolutely right that $\delta$ need not vanish with $n$: it is a property of the black-box prior. The point of the theory is precisely to characterize the finite-sample tradeoff between black-box error ($\delta^2$) and the rate of learning the residual from labeled data. We will state this non-asymptotic viewpoint much more explicitly.
>
> **On "minimax optimality" wording (Weakness 3):**
> We agree that our current wording is too loose. The correct statement is that the estimator matches the leading minimax term up to a lower-order additive validation/selection cost $O(1/n)$, rather than that the upper and lower bounds coincide exactly. We will revise the wording throughout.
>
> **On the relationship among Alg 1 / Thm 5.5 / Thm 5.7 (Weakness 2 & 4):**
> We will clarify the roles of the different results: Algorithm 1 is the concrete holdout-selected procedure. The upper-bound theorem analyzes a validation-selected estimator of this form and shows adaptation to the unknown prior quality. The oracle-style theorem establishes that the selected predictor performs, up to a lower-order term, as well as the better of the black-box route and the residual route.
>
> Regarding the notation: we acknowledge that the current draft uses several names and symbols to refer to related but distinct objects (e.g., "BB-Res," "Safe Residual," "$\hat{f}\_{final}$,"  "$\hat{f}\_{\mathrm{Alg1}}$"), which creates unnecessary confusion. In the revision, we will unify the terminology: Algorithm 1 will consistently be called the "Safe Residual Estimator," its output will be denoted $\hat{f}_{\mathrm{safe}}$ throughout, and each theorem statement will explicitly reference this notation. We will also add a notation table in the appendix for quick reference.
>
> **On classification experiments (Key Question 2):**
> > *Does this mean you have complete proofs for this version of algorithm for the corresponding theorems?*
>
> No, we do not claim that we currently have complete proofs for the cross-entropy / accuracy-based version. The formal results are for regression with squared loss. The vision/NLP experiments are included as practice-facing evidence that the residual-correction intuition and the phase-transition picture remain informative beyond the idealized regression setup; we will make this limitation explicit.
>
> **On Figure 2 (Key Question 3):**
> > *How does the proposed method go beyond both Zero-shot and Scratch, if it only switches between these two?*
>
> The method is not choosing between zero-shot and scratch. It chooses between $f_0$ and $f_0 + \hat r$. Since $\hat r$ is trained as a correction to $f_0$, the resulting predictor can outperform both the black-box prior and a scratch estimator. We will explain this more explicitly in the revision.
>
> Your close attention to these details has significantly improved the clarity of the paper.

---

> > ### Author Rebuttal · Reviewer_GSQs · 2026-04-01
> >
> > Thank you for the clarifications.
> >
> > Most of my concerns (Weaknesses 1-4 and Key Question 2) and misunderstanding (Key Question 3) have been addressed. I just have a small follow-up comment:
> >
> > > We will state this non-asymptotic viewpoint much more explicitly.
> >
> > If it's not too long, I would like to know how the authors will modify the statement.
> >
> > (By the way, there are some LaTeX errors in lines 782 and 976.)

---

> > > ### Author Response · Authors · 2026-04-02
> > >
> > > Thank you for the follow-up.
> > >
> > > **On the non-asymptotic statement:** Currently, Theorem 5.5 reads as a bound without explicitly flagging that all quantities are finite-sample. In the revision, we will add a preamble such as:
> > >
> > > *"The following bounds are non-asymptotic: they hold for every finite $n$, every $\delta \geq 0$, and every fixed black-box predictor $f_0$. In particular, $\delta$ is a property of the black-box prior and does not vary with $n$."*
> > >
> > > We will also revise the theorem statement to emphasize that $\sup_{f^* \in \mathcal{F}(\delta)} \mathbb{E}\|\hat{f} - f^*\|^2_{L_2(P_X)} \leq C \cdot \min\{\delta^2, n^{-2\beta/(2\beta+d)}\} + C'/n$ holds for all $n \geq 1$ with explicit constants $C, C'$ depending only on $(\beta, L, d, \sigma, B, B_0, c, C)$. This makes clear that the result is a finite-sample risk bound rather than an asymptotic rate statement, and that the phase transition at $\delta_c(n) \asymp n^{-\beta/(2\beta+d)}$ is a non-asymptotic phenomenon observable at any sample size.
> > >
> > > Thank you for prompting this clarification—it will make the theoretical contribution more precise. We will also fix the LaTeX errors at lines 782 and 976.

---

### Official Review · Reviewer_E8GA · 2026-03-13

**Soundness:** 3
**Presentation:** 3
**Significance:** 3
**Originality:** 3
**Overall Recommendation:** 4
**Confidence:** 2

**Summary:**

This paper studies black-box assisted nonparametric regression, where a fixed predictor is available and the target function is assumed to lie within $\delta$ distance from the base predictor in the function space. The main contribution is a minimax characterization and associated phase transition. The paper also proposes a simple residual-learning algorithm with holdout-based safe selection between using the black-box and a learned residual. The paper also provides synthetic experiments and downstream vision/NLP experiments intended to support the theory and the practical usefulness of the method.

**Compliance With Llm Reviewing Policy:**

Affirmed.

**Final Justification:**

The paper combines a clear and rigorous theoretical characterization with a simple, practical residual-based method that is intuitively motivated and designed to guard against negative transfer. My clarification questions have been satisfactorily addressed by the authors, so I am maintaining the score accordingly.

**Key Questions For Authors:**

Can the authors better justify the jump from regression theory to classification experiments, beyond the logit-regression analogy? In particular, why should the same minimax phase-transition picture be expected under cross-entropy training and accuracy-based selection?

**Limitations:**

Yes

**Strengths And Weaknesses:**

Strengths
- The paper formulates a rigorous statistical problem and characterizes clear minimax lower bounds, with a simple phase-transition characterization that is easy to understand.
- The proposed method is simple and practically appealing: learn a residual and then use validation-based safe selection to revert to the black-box when the residual hurts. This can be a reasonable safeguard against negative transfer.
- Theoretical motivation of the core idea is reasonably intuitive. In particular, the discussion contrasting residual correction with weighted ensembling gives some geometric intuition for why a residual parameterization may help.

Weaknesses
- The empirical comparison is not fully convincing because the baseline set is somewhat narrow. In particular, stronger adaptation or black-box transfer baselines, such as Gu 2023 and Lee 2025, seem to be relevant to the stated problem setting.
- Some claims are overstated. For example, the statement that pseudo-labeling/self-training “lack rigorous theoretical grounding regarding sample efficiency” seems too broad, since there is prior theoretical work on self-training under distribution shift/subpopulation shift (such as Cai 2021 and Joo 2024).


Gu 2023. Few-shot continual infomax learning. CVPR

Lee 2025. Tripartite Weight-Space Ensemble for Few-Shot Class-Incremental Learning. CVPR

Cai 2021. A Theory of Label Propagation for Subpopulation Shift. ICML

Joo 2024. Improving self-training under distribution shifts via anchored confidence with theoretical guarantees. NeurIPS

---

> ### Author Rebuttal · Authors · 2026-03-24
>
> Thank you for the careful review and for highlighting both the strengths of the paper and the places where the scope and positioning should be stated more precisely. We appreciate your positive assessment of the problem formulation, the phase-transition characterization, and the practical appeal of safe residual correction.
>
> **On the related-work wording regarding self-training:**
> > *The statement that pseudo-labeling/self-training “lack rigorous theoretical grounding” seems too broad (e.g., Cai 2021, Joo 2024).*
>
> We agree. That phrasing is too broad in its current form. Cai (2021) and Joo (2024) do provide theoretical guarantees in distribution-shift / subpopulation-shift settings, and we will revise the text to acknowledge this explicitly. Our intended point was narrower: these works do not study the minimax black-box-assisted regression problem considered here, nor the phase transition
> \[
> \min\{\delta^2,\; n^{-2\beta/(2\beta+d)}\}
> \]
> under *fixed black-box access*. We will correct the wording and add the missing citations.
>
> **On the empirical baselines (Gu 2023, Lee 2025):**
> > *The empirical comparison is not fully convincing because stronger adaptation / black-box transfer baselines seem relevant.*
>
> We agree that the empirical section should better explain why these particular baselines were chosen. Our goal was not to claim superiority over the broadest possible transfer-learning toolbox, but to isolate the specific statistical question studied in the paper: how to exploit a *fixed black-box prior* \(f_0\) when the learner has no source-data access and no parameter-space access to the pretrained model.
>
> For that reason, the baselines were selected to correspond to the key quantities and alternatives in the theory, rather than to serve as a generic benchmark suite. Concretely:
> - **BB-only** instantiates the prior-error floor \(\delta^2\),
> - **Scratch** instantiates the standard nonparametric learning route and its rate \(n^{-2\beta/(2\beta+d)}\),
> - **Validation-tuned weighted ensembling** instantiates the linear-combination alternative analyzed geometrically in Section 4.3,
> - **Concatenation** tests a simple way of injecting black-box information without residual centering,
> - and **Residual correction** is the method predicted by the theory to recover the favorable regime adaptively.
>
> We appreciate the references to Gu (2023) and Lee (2025). These are relevant neighboring empirical directions, but they rely on materially different assumptions—e.g., continual/class-incremental settings or parameter/weight-space adaptation of the pretrained model—whereas our paper studies a fixed function-access interface. We will make this distinction explicit in the revision and clarify that the current experiments are designed to validate the theoretical comparison under the paper’s access model, rather than to claim coverage of all transfer/adaptation baselines.
>
> **On regression theory versus classification experiments:**
> > *Why should the same minimax phase-transition picture be expected under cross-entropy training and accuracy-based selection?*
>
> We agree that this point should be stated much more carefully. Our formal guarantees are for regression with squared loss. The vision/NLP experiments are **not** claimed as theorem-level extensions of Theorems 5.4–5.7 to cross-entropy training and accuracy-based selection. We will revise the wording accordingly, replacing phrases such as “validate the theory” with a more precise statement: these experiments provide practice-facing evidence that the same qualitative tradeoff appears beyond the idealized regression setting.
>
> The reason we expect the same qualitative picture is structural rather than theoremic. The core mechanism in our analysis is the tradeoff between prior quality and the complexity of estimating the correction from limited labeled data. That mechanism is not unique to squared loss. In classification, when one works at the level of score/logit functions, the excess cross-entropy risk can be expressed through a KL-type discrepancy between the true class-probability vector and the predicted one. If the black-box scores are already close to the target scores, only a small correction is needed and the problem is prior-dominated; if they are not, the learner must estimate a larger correction from data and faces the same sample-complexity barrier. We will make clear that this is an intuition/structural parallel, not a formal minimax theorem for classification.
>
> We are grateful for your balanced assessment and for the specific references. They have helped us sharpen both the positioning and the scope of our claims.

---

> > ### Author Rebuttal · Reviewer_E8GA · 2026-04-03
> >
> > Thanks for addressing my concerns. I'll maintain my score.

---

> > > ### Author Response · Authors · 2026-04-03
> > >
> > > Thank you for confirming and for the constructive review. The references you suggested (Cai 2021, Joo 2024) and the question on the regression-to-classification gap have been particularly valuable in helping us sharpen the scope of our claims. The revision will reflect these improvements.

---

### Official Review · Reviewer_z3Sh · 2026-03-13

**Soundness:** 4
**Presentation:** 3
**Significance:** 3
**Originality:** 3
**Overall Recommendation:** 5
**Confidence:** 3

**Summary:**

In this paper, the authors consider the problem of nonparametric regression based on black-box estimator. It means that, for a function $f\^\*$ needed to estimate, given that we only have access to a black box estimator $f_0$, we should estimate the true $f^*$ based on  samples. The main contribution of this paper can be summarised as follows:

(R1) They provide an algorithm for the black-box estimator (Algorithm 1).

(R2) They give a rigorous theoretical analysis for this problem, including the upper bound and minimax lower bound of proposed estimation, based on sample size and error of black-box estimator.

(R3) They conduct several experiments in real dataset to justify their theoretical finding (CIFAR 100, CLIP,...).

**Compliance With Llm Reviewing Policy:**

Affirmed.

**Key Questions For Authors:**

(Q1) Can we release the condition about the covariate, including the its support lies in $[0,1]^d$, and the lower bound of density function?

(Q2) Can we release the condition about regularity of black-box predictor, i.e. we may suppose that the predictor lies rather on an even more general space, such as $L_2$?

(Q3) The authors suggest many applications of their methods in NLP, Computer Vision, LLM. Does this idea have any implication in generative AI, or Diffusion Model?

**Limitations:**

yes

**Strengths And Weaknesses:**

- Strength:

(S1) The theoretical analysis is comprehensive and gives an interesting inside about the black-box estimation. In particular, they discover the phase transition phenomenon, which suggests a possible balance between black-box error and sample size. The proof sounds rigorous and correct.

(S2) The experiment is conducted in real dataset, which strongly support their theoretical finding.

(S3) The authors give an interesting justification about the reason why linear ensembling may cause potential issue, which based on Hilbert space analysis.

(S4) The paper is well-written and easy to follow.

- Weakness:

(W1) Assumption about compactness for the distribution of covariate $P_X$ somewhat limits the applicability of the model. In fact, in some real setting, the range of the dataset is not controllable.

(W2) The factor $n^{-1}$ suggests a discrepancy between the minimax lower bound and the upper bound. In fact, if the error of black-box model is less than $n^{-1/2}$, theorem 5.5 still gives a upper bound of $n^{-1}$. Can we achieve a better estimation for the upper bound?

- Minor suggestion: Some references for [Tsybakov 2009] in Appendix has some errors.

---

> ### Author Rebuttal · Authors · 2026-03-24
>
> Thank you for the positive assessment and for the helpful technical questions. We are glad that the main theoretical contribution—the minimax characterization and the phase-transition picture—came through clearly.
>
> **On the design assumptions (W1, Q1):**
> > *Assumption about compactness for the distribution of covariate $P_X$ somewhat limits the applicability... Can we release the condition?*
>
> Yes, the compact-support / density lower-and-upper-bound conditions are standard random-design assumptions, and we should explain their role more explicitly. They are used to relate population $L_2(P_X)$ risk to Lebesgue $L_2$ risk and to make the lower/upper bound arguments technically clean. The support $[0,1]^d$ should be viewed as a normalized bounded-domain assumption rather than as a substantive application claim. Relaxing bounded support or weakening density assumptions is important future work, but is beyond the present theorem statements.
>
> **On regularity of the black-box predictor (Q2):**
> > *Can we release the condition about regularity of black-box predictor?*
>
> An important clarification is that the paper does not assume the black-box predictor $f_0$ itself lies in a Holder/Sobolev class. The smoothness assumption is placed on the residual $r^* = f^* - f_0$, together with an $L_2(P_X)$ radius constraint $\|r^*\| \le \delta$. Thus the analysis already allows $f_0$ to come from a much more general source; what matters is that the correction around $f_0$ is regular.
>
> **On the $O(1/n)$ discrepancy (W2):**
> > *The factor $n^{-1}$ suggests a discrepancy between the minimax lower bound and the upper bound... Can we achieve a better estimation?*
>
> We agree that the current wording should be more precise. Our intended claim is not exact equality between the upper and lower bounds, but matching of the leading minimax term up to a lower-order additive validation/selection cost. Since $2\beta/(2\beta+d) < 1$ for finite $d$, this $O(1/n)$ term is asymptotically smaller than the nonparametric leading term. We will revise "minimax optimality" to make this precise.
>
> Regarding whether the $O(1/n)$ term can be removed entirely: this term arises from the concentration inequality used in the holdout-based model selection step (Lemma D.5 / Theorem 5.7). It reflects the finite-sample cost of distinguishing between two candidate predictors using $n_{\mathrm{val}}$ validation points. In the prior-dominated regime where $\delta^2 \ll n^{-2\beta/(2\beta+d)}$, this $O(1/n)$ term is indeed the binding constraint. Closing this gap would likely require either a different selection mechanism (e.g., Lepski-type adaptation that avoids sample splitting) or a refined concentration analysis. We view this as an interesting open problem and will flag it explicitly in the revision.
>
> **On generative AI / diffusion models (Q3):**
> We view this as a promising future direction rather than a present claim. At a high level, one could imagine applying the same "black-box prior + safe residual correction" principle to score or denoising function estimation, but our current results do not establish this and we will keep the discussion clearly at the level of future work.
>
> We will also fix the bibliography issue you noted in the appendix. Your technical scrutiny and thoughtful suggestions have made this a much stronger paper.

---

> > ### Author Rebuttal · Reviewer_z3Sh · 2026-04-03
> >
> > I think that all my theoretical concerns have been resolved adequately. In addition, after reading your rebuttal, I realize that some technical assumption are universal in statistics/ML. Thus, I decide to keep my score.

---

> > > ### Author Response · Authors · 2026-04-03
> > >
> > > Thank you for confirming that the concerns have been resolved, and for the thoughtful engagement throughout the review process. We are glad the clarifications on the technical assumptions were helpful. Your questions—particularly regarding the $O(1/n)$ gap and the regularity conditions—have sharpened our understanding of how to present these results, and the revision will reflect this. We appreciate your support of this work.

---

### Decision · Program_Chairs · 2026-04-30

**Decision:**

Accept (regular)

**Comment:**

This paper studies nonparametric regression using a black-box estimator, treating foundation models as priors and formulating a minimax optimization problem.

The reviewers were uniformly enthusiastic about this paper, praising the theoretical analysis, including the minimax lower bound, clarity of exposition, and interesting discussions. The authors conducted convincing experiments on real datasets using downstream applications with computer vision and NLP models that confirm theoretical claims.

I recommend acceptance.